# Learning to Simulate Self-Driven Particles System with Coordinated Policy Optimization

**Zhenghao Peng[⋆], Quanyi Li[§], Ka Ming Hui[⋆], Chunxiao Liu[†], Bolei Zhou[⋆]**
[⋆]The Chinese University of Hong Kong, [†]SenseTime Research
[§]Centre for Perceptual and Interactive Intelligence

## Abstract

Self-Driven Particles (SDP) describe a category of multi-agent systems common in everyday life, such as flocking birds and traffic flows. In a SDP system, each agent pursues its own goal and constantly changes its cooperative or competitive behaviors with its nearby agents. Manually designing the controllers for such SDP system is time-consuming, while the resulting emergent behaviors are often not realistic nor generalizable. Thus the realistic simulation of SDP systems remains challenging. Reinforcement learning provides an appealing alternative for automating the development of the controller for SDP. However, previous multi-agent reinforcement learning (MARL) methods define the agents to be teammates or enemies before hand, which fail to capture the essence of SDP where the role of each agent varies to be cooperative or competitive even within one episode. To simulate SDP with MARL, a key challenge is to coordinate agents' behaviors while still maximizing individual objectives. Taking traffic simulation as the testing bed, in this work we develop a novel MARL method called Coordinated Policy Optimization (CoPO), which incorporates social psychology principle to learn neural controller for SDP. Experiments show that the proposed method can achieve superior performance compared to MARL baselines in various metrics. Noticeably the trained vehicles exhibit complex and diverse social behaviors that improve performance and safety of the population as a whole. Demo video and source code are available at: `https://decisionforce.github.io/CoPO/`.

## 1 Introduction

Self-Driven Particles (SDP) describe a wide range of multi-agent systems (MAS) in nature and human society. In SDP, individual agent pursues its own goal and interacts with each other following simple local alignment, and then the population exhibits complex collective behaviors [47]. We commonly see the phenomena of collective behaviors, such as the flocking birds [4], molecular motors [6], human crowd [17, 16], and the traffic system [20]. To understand and simulate such phenomena, researchers have developed a number of SDP models. For example, simple-rule based models [7] or Hydrodynamic equations based models [2, 19] can simulate the SDP very well in an unconstrained environment with random movement and resemble complex behaviors such as schooling fish [12] and marching locusts [5]. However, in a more structured environment such as a particular traffic scene where the interactions of agents are time-varying and the environment is non-stationary, it is difficult to design manual controllers or use rules to recover the underlying collective behaviors.

When the interactive environment is available, reinforcement learning becomes a promising approach to learn the controllers for actuating the SDP. Recently, many multi-agent reinforcement learning (MARL) methods have been developed to play competitive multi-player games, such as Hide and Seek [1], Football [23], Go and other board games [34], and StarCraft [33]. However, it is challenging to apply the existing MARL to simulate SDP systems. One essential issue is that each

35th Conference on Neural Information Processing Systems (NeurIPS 2021).

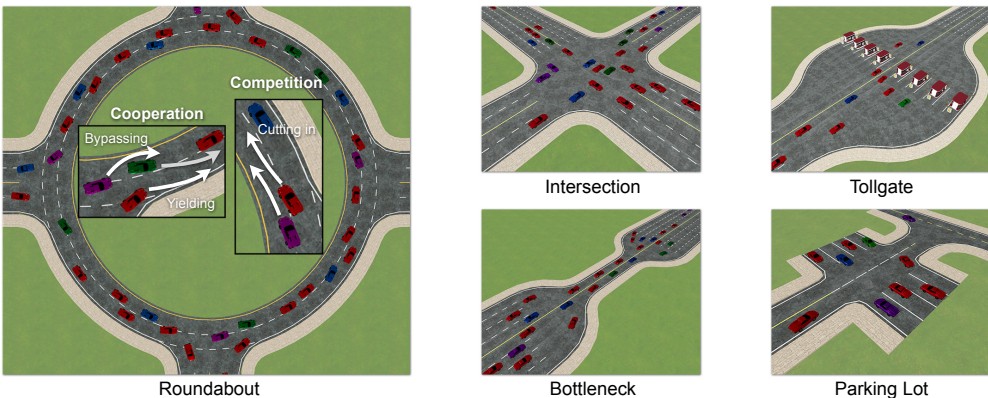

Figure 1: The five environments used in our evaluation. We highlight two cooperative and competitive events in the vehicle interactions.

constituent agent in a SDP system is self-interested and the relationship between agents is constantly changing. As illustrated in Fig. 1**A**, the vehicles in the traffic system demonstrate social behaviors that are either cooperative or competitive depending on the situation. Meanwhile, the cooperation and the competition naturally emerge as the outcomes of the self-driven instinct and the multi-agent interaction. Thus it is difficult to generate those social behaviors through a top-down design. Furthermore, most of the MARL methods assume that the role of each agent is pre-determined and fixed within one episode. In cooperative MARL methods [32, 41, 9, 8], all agents need to cooperate to maximize a joint scalar reward. However, the credit decomposition problem centered in such setting is not the major issue of simulating SDP systems, because each agent in SDP has its individual objective and the credit decomposition is no longer needed. On the other hand, directly applying independent learning [43, 35] will lead to a group of agents with unreasonable behaviors that are aggressive or egocentric, since each agent is trained to maximize its own reward.

In order to simulate SDP systems, the coordination of the self-interested constituents is the major issue. The coordination of agents in MARL has been previously studied in the context of mixed cooperative and competitive tasks [25, 38]. However, seldom works focus on the simulation of SDP systems. Taking the microscopic traffic simulation as an example, existing works focus on learning high-level controller to indirectly actuate vehicles [48], or using independent learner to train policies executing in the environments engaging a limited set of agents on simple scenes [31, 56], without considering the coordination problem. Instead, we aim to control all the vehicles in the scene by operating on the low-level continuous control space. This allows a higher degree of freedom to learn diverse behaviors but poses a challenging continuous control problem. In this work we propose a novel MARL method called **Co**ordinated **P**olicy **O**ptimization (**CoPO**) to facilitate the coordination of agents at both local and global levels. To evaluate the proposed method, as illustrated in Fig. 1, we construct five typical traffic environments as the testing bed. These environments contain rich interactions between agents and complex road structures. Besides, we develop three task-agnostic metrics to characterize different aspects of the learned populations. We show that the proposed CoPO method can achieve superior performance compared to the baselines. Noticeably, collective social behaviors that resemble real-world patterns emerge in the trained population.

## 2   Related Work

**Multi-Agent Reinforcement Learning**. Three typical task settings are explored in MARL [18]: the fully cooperative task, the fully competitive task, and the mixed cooperative and competitive task. Many works focus on the problems in cooperative tasks such as credit assignment [32, 41, 9, 8, 55] and communication [10, 46]. In competitive tasks, the challenges include modeling the opponents [11] and generating meaningful opponent via self-play [15, 49, 1]. In mixed cooperative and competitive task [29, 25, 38], the roles of the agents are mostly predefined and fixed at run-time. Instead, in this work we hope to simulate SDP systems without imposing any constraint on the roles of the agents thus there is more space for emergent social behaviors. Apart from the fixed role problem, previous

methods like MADDPG [29], QMIX [32], QTRAN [40], and COMA [9] are only applicable to discrete action space. We instead focus on learning policies for continuous control.

The independent PPO [35] and the Mean Field MARL [52] are two recent methods applied for continuous control in MARL. The Mean Field MARL method proposed by Yang *et al.* [52] approximates the value of global reward $Q(s, \mathbf{a})$, $\mathbf{a} = [a_1, ..., a_N]$ by agent-independent value functions $Q_i(s, a_i, \bar{a}_i^{\mathrm{N}})$, where $\bar{a}_i^{\mathrm{N}}$ is the average action of the nearby agents around agent $i$. Following similar design but in the context of simulating SDP, we use the mean field to compute neighborhood reward and use that reward to enforce local coordination, instead of using mean field to factorize joint value via averaging actions as done in [52].

**Traffic simulation**. Simulating the macroscopic and microscopic traffic systems has been studied for decades [21]. As for the macroscopic traffic system where the dynamics of individuals is simple and the high-level characters of flow such as the traffic throughput are more concerned, early works focus on designing controllers to direct a fleet of vehicles into a dense and stable platoon [39, 3]. CityFlow [53] and FLOW [51] use RL agents to steer the low-level controllers of vehicles in order to investigate the traffic phenomenon in large-scale macroscopic traffic flow. For microscopic system, researchers also use behavioral model [45] or hand-designed control laws [22, 50] to control individual vehicles. Zhou *et al.* [56] propose SMARTS simulator to investigate the interaction of RL agents and social vehicles in atomic traffic scenes. Pal *et al.* [31] find the emergent road rules of MARL agents trained by simple independent learner, which are also shown in our experiments. In this work, we work on the traffic environments with complex real-world scenarios such as tollgate and parking lot. The traffic flow can contain more than $40$ vehicles. In contrast to previous work, we also propose novel method to improve the collective motions that better simulate SDP systems.

## 3 Preliminaries

### 3.1 Problem Setting

SDP system has two distinguishable properties: (1) each individual agent is self-interested, (2) the individual agent can vary to be cooperative or competitive with others within one episode and (3) the individual agent only actively interacts with its nearby agents.

To accommodate these properties, we formulate a SDP system as a set of Decentralized Partially Observable Markov Decision Processes (Dec-POMDPs) [14] represented by a tuple $G = <\mathcal{E}, \mathcal{S}, \mathcal{A}, \mathcal{P}, \mathcal{R}, \rho_0, \mathcal{O}, \mathcal{Z}, \gamma>$. $\mathcal{E}$ is the set of the agent indices. We define the active agent set at each environmental time step $t$ as $\mathcal{E}_t = \{i_{1,t}, ..., i_{k_t,t}\} \subset \mathcal{E}$ wherein $k_t$ is the number of existing agents at this step. We consider a partially observable setting in which each agent can not access the environmental state $s_t$ and instead draws individual observation $o_{i,t}$ according to the local observation function $o_{i,t} = \mathcal{Z}_i(s_t) : \mathcal{S} \to \mathcal{O}$. The joint action space $\mathcal{A}_t \subset \mathcal{A}$ is composed of the union of all active agents' action spaces $\bigcup_{i=1}^{k_t} \mathcal{A}_{i,t}$. Each agent $i$ chooses its action $a_{i,t} \in \mathcal{A}_{i,t}$ following its policy $a_{i,t} \sim \pi_i(\cdot|o_t)$. These actions form a joint action $\mathbf{a}_t = \{a_{i,t}\}_{i=1}^{k_t} \in \mathcal{A}_t$ causing a transition on the environmental state subjected to the state transition distribution $\mathcal{P}(s_{t+1}|s_t, \mathbf{a}_t)$. Each agent receives their *individual reward* from its reward function $r_{i,t} = \mathcal{R}_i(s_t, \mathbf{a}_t)$. $\rho_0$ is the initial state distribution and $\gamma$ is the discount factor. So in test time our system is decentralized, where each agent is fed with its local observation and actuated by its policy and interacts with others.

Here we define the *environmental episode* $\tau = \{(s_t, \mathbf{a}_t, r_{1,t}, ..., r_{k_t,t})\}_{t=0}^{T}$, where $T$ is the environmental horizon. An environmental episode thus contains a set of *agent episodes*: $\{(o_{i,t}, a_{i,t}, r_{i,t})\}_{t=t_i^s}^{t_i^e}$, wherein $t_i^s$ and $t_i^e$ denote the environmental steps when agent $i$ enters and terminates, respectively. Suppose the policy $\pi_i$ is parameterized by $\theta_i$, the *individual objective* for each agent is defined as the discounted sum of individual reward (namely the individual return): $J_i^I(\theta_i) = \mathbb{E}_\tau[R_i(\tau)]$, wherein $R_i(\tau) = \sum_{t=t_i^s}^{t_i^e} \gamma^{t-t_i^s} r_{i,t}$.

### 3.2 Individual Policy Optimization

To simulate SDP systems, a simple approach is the independent policy optimization (IPO) [35] where each agent maximizes individual objective as if in the single-agent environment. Denote the individual value function is $V_{i,t}^I = V_i^I(s_t) = \mathbb{E}[\sum_{t'=t}^{t_i^e} \gamma^{(t'-t)} r_{i,t}]$ and the corresponding advantage

function is $A^I_{i,t} = A^I_i(s_t, a_{i,t}, \mathbf{a}_{\neg i,t}) = r_{i,t} + \gamma V^I_i(s_{t+1}) - V^I_i(s_t)$, wherein $\mathbf{a}_{\neg i,t} = \{a_i\}^{k_t}_{j=1, j\neq i}$ is the shorthand of other agents' actions, the policy gradient method computes the gradient of individual objective as [42]:

$$\nabla_{\theta_i} J^I_i(\theta_i) = \underset{(s,\mathbf{a})}{\mathbb{E}} [\nabla_{\theta_i} \log \pi_{\theta_i}(a_i|s) A^I_i(s, a_i, \mathbf{a}_{\neg i})]. \tag{1}$$

As a common practice, in PPO algorithm [37] the clipped importance sampling factor $\rho = \pi_{i,\text{new}}(a_i|s)/\pi_{i,\text{old}}(a_i|s)$ is used to mitigate the distribution shift occurred after the policy is updated for several epochs, wherein $\pi_{i,\text{old}}$ is the behavior policy that generate samples and $\pi_{i,\text{new}}$ is the latest policy parameterized by $\theta_i$. We call the resulting objective as the *surrogate objective*:

$$\hat{J}^I_i(\theta_i) = \underset{(s,\mathbf{a})}{\mathbb{E}} \min(\rho A^I_i, \text{clip}(\rho, 1 - \epsilon, 1 + \epsilon) A^I_i). \tag{2}$$

$\epsilon$ is hyper-parameter. By conducting the stochastic gradient ascent on the surrogate objective w.r.t. the policy parameters, the expected individual return can be improved. In continuous control RL problem, both the state space and action space are high-dimensional. Therefore $V^I_i$ is approximated using a neural network whose input is typically the local observation. In the centralized training and decentralized execution (CTDE) framework [29], the value function takes global information as input by concatenating all agents' local observations.

## 4 Coordinated Policy Optimization

In a SDP system each agent has its individual objective. However, if we simply maximize each individual reward, the system will have sub-optimal solutions where, for example, the agents will become aggressive and egocentric, jeopardizing the performance of the population and leading to critical failures. On the contrary, if we apply cooperative learning schemes [32, 41] and consider the summation of individual reward as the joint objective, the trained agents will exhibit unreasonable behaviors such as sacrificing oneself to improve group reward, which is not expected in SDP systems. To find a balance, as illustrated in Fig. 2 and Algorithm in the Appendix, we devise a novel MARL algorithm called **Co**ordinated **P**olicy **O**ptimization (**CoPO**) to facilitate the bi-level coordination of agents to learn the controllers of the SDP systems. Following the CTDE framework, during centralized training, we first propose individual learning objectives by local coordination (Sec. 4.1), a mechanism inspired by the fact that each individual agent is mostly affected by its nearby agents [52]. We coordinate agents' objectives in neighborhood by mixing rewards following a common social psychological principle. We further design the meta-learning technique to optimize the local coordination process, leading to the global coordination (Sec. 4.2) which can improve collective performance of the population.

### 4.1 Local Coordination

IPO only maximizes the individual objective (Eq. 1) which leads to egocentric policies that damage the overall performance of the population. However, in natural SDP systems, e.g. animal groups, there exists a certain level of cooperative behavior that not only benefits others, but also results better individual utility. To resemble such feature, we incorporate the ring measure of social value orientation [28, 38], a common metric from social psychology, into our formulation of local coordination. Concretely, we define the *Local Coordination Factor* (LCF) as a degree $\phi \in [-90°, 90°]$ describing an agent's preference of being selfish, cooperative, or competitive.[1] We then define the *neighborhood reward* as:

$$r^N_{i,t} = \frac{\sum_{j \in \mathcal{N}_{d_n}(i,t)} r_{j,t}}{|\mathcal{N}_{d_n}(i,t)|}, \text{ wherein } \mathcal{N}_{d_n}(i,t) = \{j : ||\text{Pos}(i) - \text{Pos}(j)|| \leq d_n\}. \tag{3}$$

$\mathcal{N}_{d_n}(i,t)$ defines the neighborhood of agent $i$ within the radius $d_n$ at step $t$. Note that the neighborhood of agent is time-varying, therefore the the neighborhood reward can not be simply computed from a fixed set of agents across the episode. Weighted by LCF, the *coordinated reward* is defined as:

$$r^C_{i,t} = \cos(\phi) r_{i,t} + \sin(\phi) r^N_{i,t}. \tag{4}$$

---

[1] The social value orientation can be in range $[-180°, 180°]$. We only take the meaningful half of the range.

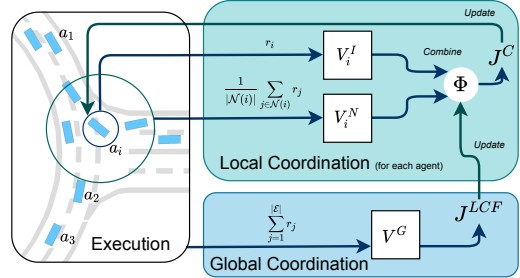
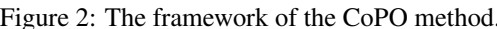

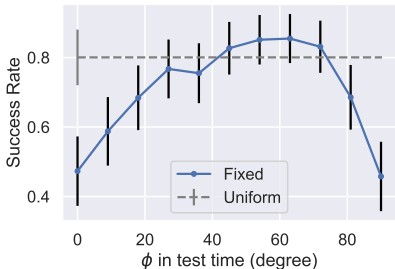

Figure 2: The framework of the CoPO method.

Figure 3: The test-time performance of a population trained with uniformly sampled LCF.

The LCF enables the local coordination in the optimization process. Supposing the agents maximize the coordinated reward with $\phi = 0°, 90°, -90°$, the agents should become egoistic, altruistic, or sadistic respectively. Illustratively, Fig. 3 demonstrates an interesting preliminary result on a simple version of Roundabout environment. During IPO training, we randomly select $\phi \sim \mathcal{U}(0°, 90°)$ and assign it to each agent in each episode. Current $\phi$ is fed to policy as extra observation. In test time, we run the trained population for multiple runs. In each run we set a fixed LCF for all agents. The test results show that a peak of success rate at certain LCF at test time. This suggests that increasing the awareness of neighbors' interests when training individual agents will propose better policies that increase the collective performance. IPO might lead to sub-optimal policies since the individual objectives never consider local coordination with other agents.

In order to incorporate the LCF into training process to improve the population performance, apart from the value function approximator for the individual reward (Sec. 3.2), we use another value function to approximate the discounted sum of the neighborhood reward $V_{i,t}^N = \mathbb{E}[\sum_t^{t_i^e} \gamma^{t-t_i^s} r_{i,t}^N]$ and the neighborhood advantage is therefore calculated as: $A_{i,t}^N = r_{i,t}^N + \gamma V_{i,t+1}^N - V_{i,t}^N$. The coordinated advantage can be derived as: $A_{\Phi,i,t}^C = \cos(\phi)A_{i,t}^I + \sin(\phi)A_{i,t}^N$. We now have the *coordinated objective*, which takes the utilities of neighbors into consideration, as follows:

$$J_i^C(\theta_i, \Phi) = \underset{(s,\mathbf{a})}{\mathbb{E}} [\min(\rho A_{\Phi,i}^C, \text{clip}(\rho, 1-\epsilon, 1+\epsilon)A_{\Phi,i}^C)]. \tag{5}$$

We follow a similar training procedure as IPO in Sec. 3.2 with the advantage replaced by the coordinated advantage. To improve diversity, we use distributional LCF $\phi \sim \mathcal{N}(\phi_\mu, \phi_\sigma^2)$, where $\Phi = [\phi_\mu, \phi_\sigma]^\top$ are the learnable parameters of the LCF distribution.

## 4.2 Global Coordination

The major challenge of local coordination is the selection of LCF. Though the experiment in Fig. 3 shows that there exists an optimal LCF that can maximize the population success rate, it is laborious to search for such LCF. We therefore introduce global coordination which enables automatic search of the best LCF distribution to improve population efficiency.

In this section, we introduce how we accomplish global coordination with bi-level optimization method that computes a meta-gradient of the global objective. We first define the *global objective* that indicates the performance of a population:

$$J^G(\theta_1, \theta_2, ...) = \mathbb{E}_\tau[\sum_{i \in \mathcal{E}_\tau} \sum_{t=t_i^s}^{t_i^e} r_{i,t}] = \mathbb{E}_\tau[\sum_{t=0}^{T} \sum_{i \in \mathcal{E}_{\tau,t}} r_{i,t}], \tag{6}$$

wherein $\mathcal{E}_{\tau,t}$ denotes the active agents at environmental time step $t$ of the episode $\tau$. Here we remove $\gamma$ since it is ambiguous when the start steps of agents are not aligned. Eq. 6 can not be directly optimized when computing policy gradients of individual policies' parameters.

We introduce the concept of *individual global objective* to make the optimization of the global objective feasible. By this way, we can turn the system-level optimization process into agent-level optimization processes, so that the individual agent's data can be easily accessed. We factorize

$J^G(\theta_1, ...)$ to individual global objectives $J_i^G(\theta_i)$ such that maximizing the objective for each agent $i$ is equivalent to maximizing Eq. 6. The *individual global objective* is defined as:

$$J_i^G(\theta_i|\theta_1, ...) = \mathbb{E}_\tau\Big[\sum_{t=t_i^s}^{t_i^e} \frac{\sum_{j\in\mathcal{E}_{\tau,t}} r_{j,t}}{|\mathcal{E}_{\tau,t}|}\Big]. \tag{7}$$

If we introduce the *global reward* $r^G = \sum_{j\in\mathcal{E}_{\tau,t}} r_{j,t}/|\mathcal{E}_{\tau,t}|$ as the average reward of all active agents at step $t$, then this objective is the cumulative global reward in the time interval when agent $i$ is alive.

**Proposition 1** (Global objective factorization). *Suppose each agent $i \in \mathcal{E}$ can maximize its individual global objective $J_i^G$, then the original global objective $J^G$ is maximized.*

We prove that such factorization does not affect the optimality of global objective in Appendix.

To improve global objective via optimizing LCF, we need to compute the gradient of Eq. 7 w.r.t. $\Phi$. Denote the parameters of policies before and after optimizing Eq. 5 as $\theta_i^{\text{old}}$ and $\theta_i^{\text{new}}$ respectively, we can compute the gradient applying the chain rule as:

$$\nabla_\Phi J_i^G(\theta_i^{\text{new}}) = \underbrace{\nabla_{\theta_i^{\text{new}}} J_i^G(\theta_i^{\text{new}})}_{\text{1st Term}} \underbrace{\nabla_\Phi \theta_i^{\text{new}}}_{\text{2nd Term}}. \tag{8}$$

The 1st term resembles the policy gradient as Eq. 2 with the objective replaced by $J_i^G$:

$$\text{1st Term} = \mathbb{E}_{(s,\mathbf{a})\sim\theta_i^{\text{old}}}[\nabla_{\theta_i^{\text{new}}} \min(\rho A^G(s, \mathbf{a}), \text{clip}(\rho, 1-\epsilon, 1+\epsilon) A^G(s, \mathbf{a}))]. \tag{9}$$

Here we use an extra global value function $V^G$ to estimate the value of global reward $r^G$, then compute the global advantage $A^G$. The samples $(s, \mathbf{a})$ are generated by the behavior policy $\theta_i^{\text{old}}$.

By following first-order Taylor expansion, the 2nd term can be computed as follows. Note that $\nabla_{\theta_i^{\text{old}}} J_i^C(\theta_i^{\text{old}}, \Phi)$ recovers the vanilla policy gradient in Eq. 1.

$$\text{2nd Term} = \nabla_\Phi(\theta_i^{\text{old}} + \alpha\nabla_{\theta_i^{\text{old}}} J_i^C(\theta_i^{\text{old}}, \Phi)) = \alpha \mathbb{E}_{(s,\mathbf{a})\sim\theta_i^{\text{old}}}[\nabla_{\theta_i^{\text{old}}} \log \pi_{\theta_i^{\text{old}}}(a_i|s)\nabla_\Phi A_{\Phi,i}^C]. \tag{10}$$

Averaging over all agents and combining Eq. 9 and Eq. 10, we can derive the LCF objective as:

$$J^{LCF}(\Phi) = \mathbb{E}_{i,(s,\mathbf{a})\sim\theta^{\text{old}_i}}[\nabla_{\theta_i^{\text{new}}} \min(\rho A^G, \text{clip}(\rho, 1-\epsilon, 1+\epsilon) A^G)][\nabla_{\theta_i^{\text{old}}} \log \pi_{\theta_i^{\text{old}}}(a_i|s)]A_{\Phi,i}^C. \tag{11}$$

We then conduct stochastic gradient ascent on $J^{LCF}$ w.r.t. $\Phi$ with learning rate $\alpha$.

The detailed procedure of CoPO is given in Appendix. In our implementation, we follow the commonly used parameter sharing technique [44] so that all agents share the same set of neural networks, which guarantees the scalability of our method. As a summary, our method totally employs 4 neural networks: the policy network, the individual value network, the neighborhood value network, and the global value network. The input to all networks is always the local observation of the agent. Our further study (see Sec. 5.5) shows that centralized critic method, which feeds global information to the value networks, does not yield better performance and usually causes unstable training. The reason might be the time-varying input to the centralized critics, that the combined observation of nearby agents is always changing and doesn't hold a temporal consistency across the episodes.

## 5 Experiments

### 5.1 Evaluation Platform

We develop a set of 3D traffic environments to evaluate MARL methods for simulating SDP systems based on MetaDrive [26], a lightweight and efficient microscopic driving simulator[2]. The environments are developed in Panda3D [13] and Bullet Engine, which can run 80 FPS on a standard

---

[2]MetaDrive can be found at: `https://github.com/decisionforce/metadrive`.

workstation. Examples of the environments are shown in Fig. 1. The descriptions and typical settings of the five traffic environments are as follows:

**Roundabout**: A four-way roundabout with two lanes. 40 vehicles spawn during environment reset. This environment includes merge and split junctions.

**Intersection**: An unprotected four-way intersection allowing bi-directional traffic as well as U-turns. Negotiation and social behaviors are expected to solve this environment. We initialize 30 vehicles.

**Tollgate**: Tollgate includes narrow roads to spawn agents and ample space in the middle with multiple tollgates. The tollgates create static obstacles where the crashing is prohibited. We force agent to stop at the middle of tollgate for 3s. The agent will fail if they exit the tollgate before it is allowed to pass. 40 vehicles are initialized. Complex behaviors such as deceleration and queuing are expected. Additional states such as whether vehicle is in tollgate and whether the tollgate is blocked are given.

**Bottleneck**: Complementary to Tollgate, Bottleneck contains a narrow bottleneck lane in the middle that forces the vehicles to yield to others. We initialize 20 vehicles.

**Parking Lot**: A compact environment with 8 parking slots. Spawn points are scattered in both parking lots and external roads. 10 vehicles spawn initially. In this environment, we allow agents to back their cars to spare space for others. Maneuvering and yielding are the key to solve this task.

Those environments cover several scenarios in previous simulators [56, 31] but are extended with real-world scenarios rather than atomic scenes. Besides, our environments support dense traffic flow where all vehicles are controlled through the low-level continuous signals. All the environments can be further concatenated to form more complex tasks for the future study of multi-task learning.

Evaluating a population simulated by MARL is an open question. In this work, we define a set of general-purpose metrics that can characterize different aspects of the SDP systems. We choose three episodic metrics with minimal assumptions on the tasks. We evaluate the competence of population with *success rate*. It is the ratio of the number of agents that reach their goals over the total number of agents in one episode. We also evaluate the *efficiency* of the population. It indicates the difference between successes and failures in a unit of time $(N_{\mathrm{success}} - N_{\mathrm{failure}})/T$. It is possible for a population to achieve high *success rate* but has low *efficiency*, because the agents in the population are running in very low velocity. The third metric we adopt here is the *safety*. It is defined as the total number of critical failures for all agents in one episode.

## 5.2 Experiment Setting

We compare the independent policy optimization (*IPO*) method [35] using PPO [37] as the individual learners. We further encode the nearby agents' states into the input of value functions (centralized critics) following the idea of Mean Field MARL [52] and form the mean field policy optimization (*MFPO*). We examine different variants of centralized critics, but find MFPO yields best performance (see Sec. 5.5). The curriculum learning (*CL*) [30] is also a baseline, where we chunk the training into 4 phases and gradually increase initial agents in each episode from $25\%$ to $100\%$ of the target number.

We conduct experiments with the aforementioned environments and algorithms using RLLib [27], a distributed learning system which allows large-scale parallel experiments. Generally, we host 4 concurrent trials in an Nvidia GeForce RTX 2080 Ti GPU. Each trial consumes 2 CPUs with 4 parallel rollout workers. Each trial is trained over roughly 1M environmental steps, which corresponds to about 55 hours in real-time transportation system and $+2,000$ hours of individual driving experience (assume averagely 40 vehicles running concurrently). All experiments are repeated 8 times with different random seeds. Other hyper-parameters are given in Appendix.

## 5.3 Main Results

Table 1 shows the main results of all the evaluated methods. Because of the bi-level coordination in policy optimization, our CoPO achieves superior success rate in all 5 environments. Noticeably in the most difficult environment, the unprotected Intersection, our method outperforms baselines with a substantial margin. That environment requires agents to cooperate frequently in order to determine who should go fist. The populations generated from other methods cause severe congestion in the

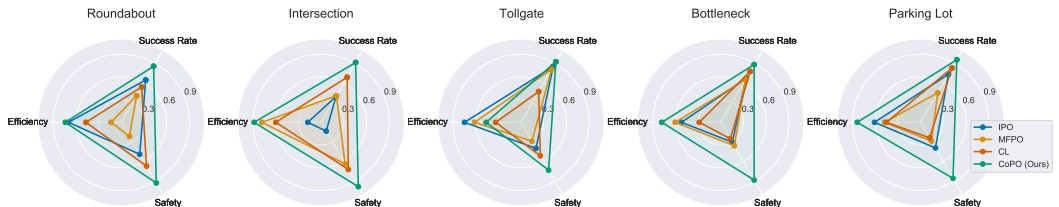

Figure 4: Performance of the trained populations from different MARL methods.

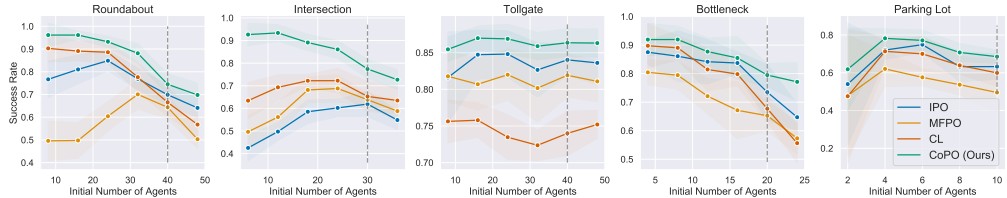

Figure 5: The success rate of different trained populations with varying number of initial agents at test time. The gray lines denote the initial number of agents in the training environments.

Table 1: Success rate of different approaches.

|  | Roundabout | Intersection | Tollgate | Bottleneck | Parking Lot |
|---|---|---|---|---|---|
| IPO | 70.81 $\pm_{1.95}$ | 60.47 $\pm_{5.79}$ | 82.90 $\pm_{2.81}$ | 72.43 $\pm_{3.79}$ | 61.05 $\pm_{2.81}$ |
| MFPO | 64.27 $\pm_{3.68}$ | 67.74 $\pm_{4.19}$ | 81.05 $\pm_{3.07}$ | 67.40 $\pm_{4.77}$ | 53.96 $\pm_{4.65}$ |
| CL | 65.48 $\pm_{3.96}$ | 62.03 $\pm_{4.41}$ | 73.72 $\pm_{3.46}$ | 68.81 $\pm_{4.39}$ | 60.62 $\pm_{2.25}$ |
| CoPO (Ours) | **73.67** $\pm_{3.71}$ | **78.97** $\pm_{4.23}$ | **86.13** $\pm_{1.76}$ | **79.68** $\pm_{2.91}$ | **65.04** $\pm_{1.59}$ |

middle of intersection which prevents any agent going through. See Sec. 5.4 and demo video for detailed visualization.

We also find the MFPO performs worse than the independent PPO in several environments. That might be due to the neighborhood changing all the time. Simply concatenating or averaging neighbors' states as the additional input to the value functions makes the training unstable. A detailed comparison of different design choices in centralized critics is given in Sec. 5.5.

Fig. 4 presents the radar plots under the three metrics. In each environment, we normalize each metric to its maximal and minimal values in all evaluated episodes for all algorithms. For *safety* measurement, we use negative values of total crashes in normalization. We can see that CoPO achieves superior results across all three metrics. Noticeably, CoPO has a substantial improvement in safety, showing the importance of the bi-level coordination.

To further evaluate the generalization of the populations, we run the trained policies in the test environments with different initial numbers of agents. As shown in Fig. 5, we find that in Roundabout and Intersection, where the interactions are extremely intensive, IPO and MFPO overfit to the initial number of agents. In the environment with sparse traffic, their performance is inferior to that in the environment with more agents. On the contrary, our method does not overfit these two environments. In Parking Lot, it seems that all methods has some degree of overfitting to the initial number of agents. The results are consistent with the finding in [24], showing that strategies learned via a single instance of MARL algorithm can overfit to the policies of other agent in the population. In Appendix, we conduct an experiment using a simple heuristic to control part of the vehicles in the population to test the generalizability of different training methods. The experiment shows that CoPO trained agents are robuster than IPO trained ones when interacting with vehicles controlled by unseen policies. It is left to future work on improving the generalization of MARL methods.

## 5.4 Behavioral Analysis

Fig. 6 plots the visualization of all trajectories occurred in one environmental episode, where the agents spawning at same entries share the same colors. The population trained from CoPO exhibits collective behaviors, while the critical failures reduce drastically compared to other populations. The

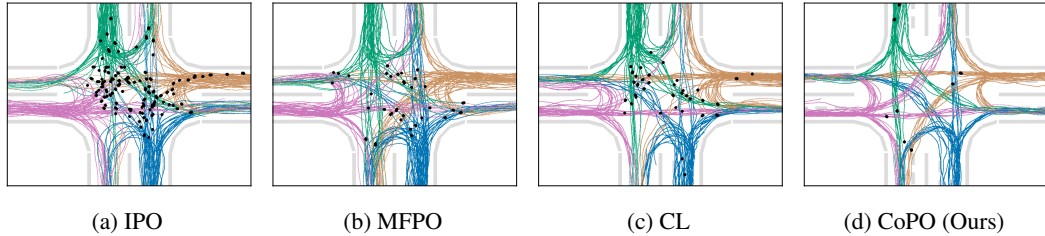

| (a) IPO | (b) MFPO | (c) CL | (d) CoPO (Ours) |

Figure 6: Plot of trajectories of trained populations in Intersection. The dark spots indicate the locations where critical failures happen. CoPO produces safer and more coordinated traffic flow.

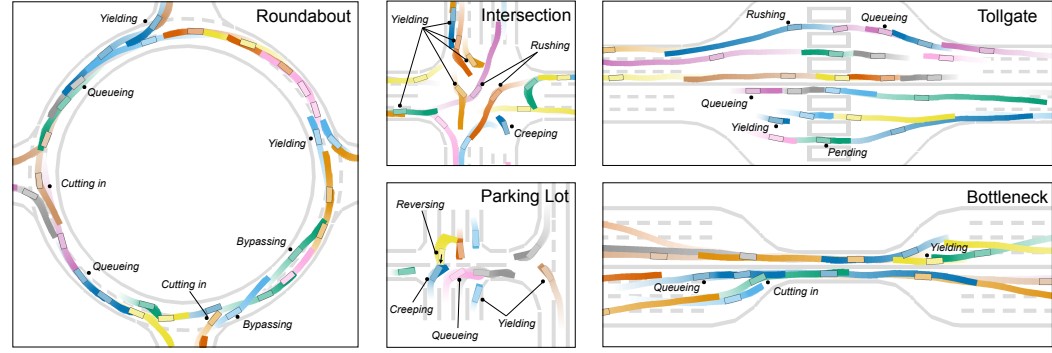

Figure 7: Visualizing the social behaviors of CoPO populations. Diverse behaviors emerge in CoPO populations, which are highlighted with black dots in the plot. We plot the past 25 and future 25 positions of each vehicle with different colors, where the luminosity of a trajectory decreases from light to dark, representing its past to future positions.

population trained from other methods tend to directly rush into the center of Intersection, causing severe congestion and frequent crashes. On the contrary, the population from CoPO tends to drive collectively and successfully avoid crashes through social behaviors such as yielding, with superior success rate and high safety. The qualitative result suggests that CoPO can resemble the collective motions widely existing in SDP systems [54].

Fig. 7 further visualizes the temporal behaviors of CoPO agents in 5 environments. Each vehicle in the population tends to drive socially and react to its nearby vehicles. A set of diverse behaviors emerge through the interactions, including socially compliant behaviors such as yielding, queuing, and even reversing to leave more room to others, as well as aggressive behaviors such as cutting in and rushing. The population trained from CoPO successfully reproduces diverse interaction behaviors in traffic systems.

## 5.5 Ablation Studies

We evaluate different variants of centralized critics in Roundabout environment. The *Concat* variant refers to simply concatenating all states of nearest K=4 agents as a long vector fed to the value networks. The *Mean Field* method uses the average of the states from nearby agents within a given radius (we use 10 meters). The *CF* refers to "counterfactual", namely adding the neighbors' actions accompanied with their observations as the input to value functions. Note that the *Mean Field w/ CF* method follows the Mean Field Actor Critic proposed in [52], the difference is that we use the training pipeline of PPO [36] in order to operate in continuous action space. As illustrated in Fig. 9, we find that the design with Mean Field is more stable compared to Concat. CF also stabilizes the training. We therefore report the results of *Mean Field w/ CF* as the MFPO method in Sec. 5.3.

Fig. 8 plots the learning curves of CoPO with and without centralized critics. We find that CoPO with centralized critics yields inferior success rate and is unstable in some environments compared to the native CoPO. This might because nearby agents are always changing and doesn't hold the temporal consistency within episode.

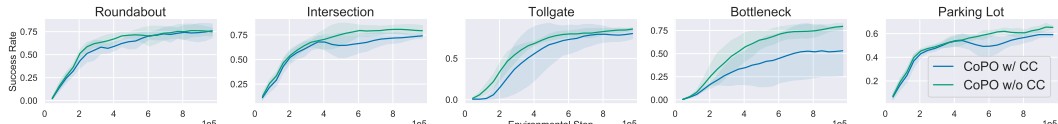

Figure 8: Comparison of CoPO with or without centralized critics.

Table 2: Ablation study on the effectiveness of global coordination. Average success rate is provided.

| Experiment | Roundabout | Intersection | Tollgate | Bottleneck | Parking Lot |
|---|---|---|---|---|---|
| **(a)** sample $\phi$ from $\mathcal{N}(0, 0.1^2)$ | $62.38_{\pm 7.26}$ | $70.23_{\pm 2.72}$ | $60.47_{\pm 5.80}$ | $71.29_{\pm 3.27}$ | $59.14_{\pm 1.29}$ |
| **(b)** maximize global reward | $0.00_{\pm 0.00}$ | $0.00_{\pm 0.00}$ | $0.00_{\pm 0.00}$ | $0.00_{\pm 0.00}$ | $0.00_{\pm 0.00}$ |
| **(c)** maximize neighborhood reward | $65.70_{\pm 2.21}$ | $66.83_{\pm 2.16}$ | $57.30_{\pm 3.46}$ | $71.62_{\pm 1.22}$ | $6.34_{\pm 1.44}$ |
| **CoPO**: sample $\phi$ from $\mathcal{N}(\phi_\mu, \phi_\sigma^2)$ | $73.67_{\pm 3.71}$ | $78.97_{\pm 4.23}$ | $86.13_{\pm 1.76}$ | $79.68_{\pm 2.91}$ | $65.04_{\pm 1.59}$ |

In Table 2**(a)**, we validate the effectiveness of global coordination by replacing the LCF distribution with a given normal distribution during the course of training. The proposed method can yield better success rate than the result of manually defined fixed LCF distributions. In **(b)**, we use IPO to maximize the global reward directly. Unfortunately, the training fails because it is hard to build the causal connection between individual's behavior with the average reward of all agents. In **(c)**, we use IPO to maximize the neighborhood reward. This method performs poorer than CoPO since we balance the individual reward and the neighborhood reward.

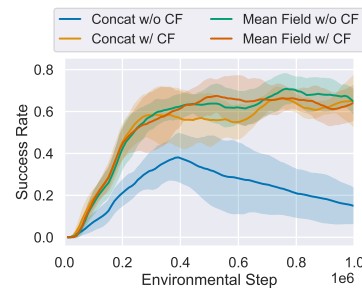

Figure 9: Comparison of design choices of centralized critic methods.

## 6 Conclusion

In this work, we develop a novel MARL method called **Co**ordinated **P**olicy **O**ptimization (CoPO) to incorporate social psychology principle to learn neural controller of SDP systems. Experiments on microscopic traffic simulation show that the proposed method can learn populations which achieve superior performance over the previous MARL methods in three general-purpose metrics. Interestingly, we find the trained vehicles exhibits complex and socially compliant behaviors that improve the efficiency and safety of the population as a whole.

## 7 Social Impact and Limitations

This work could help the development of intelligent transportation systems which would have a wide impact on society. We can analyze the emergent behaviors of the traffic under different scene structures and optimize the road structure or traffic light control to improve the traffic efficiency. Moreover, CoPO can simulate diverse traffic flow, which can be used to benchmark the generalizability of the autonomous driving systems. Besides simulating traffic flow, the proposed method is applicable for pedestrian crowd simulation to study social crowd behaviors as well as potential human stampedes and crushes.

This work focuses on decision-making problem, so we simplify the perception of the vehicles as one-channel LiDAR and assume the acquisition of accurate sensory data. In reality, the precise perception of the surroundings in self-driving remains very challenging. This work adopts 5 typical traffic scenarios in a simulator as the testbed, which are still far from emulating the complexity of real-world traffic scenes. In the future we will import real-world HD maps in the simulator to create more realistic scenarios.

## Acknowledgments and Disclosure of Funding

This project was supported by the Centre for Perceptual and Interactive Intelligence (CPII) Ltd under the Innovation and Technology Fund.

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
