# Learning to Simulate Self-Driven Particles System with Coordinated Policy Optimization Appendix

**Zhenghao Peng$^\star$, Quanyi Li$^\S$, Ka Ming Hui$^\star$, Chunxiao Liu$^\dagger$, Bolei Zhou$^\star$**
$^\star$The Chinese University of Hong Kong, $^\dagger$SenseTime Research
$^\S$Centre for Perceptual and Interactive Intelligence

## A  The procedure of CoPO

Algorithm 1 describes the overall procedure of CoPO.

---
**Algorithm 1:** The Procedure of CoPO

---
   **Input:** Maximal iterations $N$; Environmental horizon $T$; Number of training epochs in each iteration $K_p$, $K_\Phi$ for updating policy and $\Phi$, respectively; Number of mini batches in each epoch $M$.

1 Initialize environment, data buffer $D$, policy network $\pi_\theta$, individual value network $V_\eta^I$, neighborhood value network $V_\psi^N$, global value network $V_\omega^G$, LCF distribution $\mathcal{N}(\phi_\mu, \phi_\sigma^2)$.

2 **for** $I = 1, ..., N$ **do**

3     Clear buffer $D$ and store current policy $\pi_{\text{old}} \leftarrow \pi_\theta$.

4     **for** $t = 0, ..., T$ **do**

5        Assign each newly spawned agent $i$ with $\phi_i \sim \mathcal{N}(\phi_\mu, \phi_\sigma^2)$.

6        $\mathbf{a}_t = [a_{1,t}, ..., a_{k_t,t}], a_{i,t} \sim \pi_\theta(\cdot | o_{i,t}), \forall i = 1, ..., k_t$.

7        Step the environment with $\mathbf{a}_t$ and store the tuple $(s_t, \mathbf{a}_t, r_{1,t}, ..., o_{1,t}, ...)$ to $D$.

8     **for** $k = 1, ..., K_p$ **do**

9        Shuffle $D$ and slice it into $M$ batches.

10        Update $\pi_\theta$, $V_\eta^I$, $V_\psi^N$ and $V_\omega^G$ following Eq. 5 for each mini batch.

11     **for** $k = 1, ..., K_\Phi$ **do**

12        Update $\Phi$ following Eq. 11 with $D$.

---

## B  Proof of global objective factorization

The global objective $J^G(\theta_1, \theta_2, ...) = \mathbb{E}_\xi[\sum_{i \in \mathcal{E}_\xi} \sum_{t=t_i^s}^{t_i^e} r_{i,t}] = \mathbb{E}_\xi[\sum_{t=0}^{T} \sum_{i \in \mathcal{E}_{\xi,t}} r_{i,t}]$ can not be directly optimized when computing the policy gradients of individual policies' parameters. This is because when $t < t_i^s$ or $t > t_i^e$, the agent $i$ does not exist in the environment and therefore no samples $(s, a_i, \mathbf{a}_{\neg i})$ are there. Alternatively, we look for the factorization of the global objective such that optimizing it for each policy can converge to the optimal global objective.

We use the average rewards of all active agents at each time step as the individual reward for each agent. Concretely, we use the following *individual global objective*:

$$J_i^G(\theta_i | \theta_1, ...) = \mathbb{E}_\xi[\sum_{t=t_i^s}^{t_i^e} \frac{\sum_{j \in \mathcal{E}_{\xi,t}} r_{j,t}}{|\mathcal{E}_{\xi,t}|}]. \tag{1}$$

We justify our choice by the following proposition and the proof:

**Proposition 1** (Global objective factorization)**.** *Supposing each agent $i \in \mathcal{E}$ can maximize its individual global objective $J_i^G$, then the original global objective $J^G$ is maximized.*

35th Conference on Neural Information Processing Systems (NeurIPS 2021).

*Proof.* We can easily show that:

$$J^G(\theta_1, \theta_2, ...) = \mathbb{E}_{\xi}[\sum_{t=0}^{T} \sum_{i \in \mathcal{E}_{\xi,t}} r_{i,t}]$$

$$= \mathbb{E}_{\xi}[\sum_{t=0}^{T} \sum_{j \in \mathcal{E}_{\xi,t}} \frac{1}{|\mathcal{E}_{\xi,t}|} \sum_{i \in \mathcal{E}_{\xi,t}} r_{i,t}]$$

$$= \mathbb{E}_{\xi}[\sum_{t=0}^{T} \sum_{i \in \mathcal{E}_{\xi,t}} \frac{\sum_{j \in \mathcal{E}_{\xi,t}} r_{j,t}}{|\mathcal{E}_{\xi,t}|}] \quad (2)$$

$$= \mathbb{E}_{\xi}[\sum_{i=1}^{|\mathcal{E}_{\xi}|} \sum_{t=t_i^s}^{t_i^e} \frac{\sum_{j \in \mathcal{E}_{\xi,t}} r_{j,t}}{|\mathcal{E}_{\xi,t}|}]$$

$$= \sum_{i} J_i^G(\theta_i | \theta_1, ...)$$

Since $J^G(\theta_1, \theta_2, ...)$ is the summation of $J_i^G(\theta_i)$, following the idea of [11], we have:

$$\frac{\partial J^G}{\partial J_i^G} \geq 0, \forall i = 1, .... \quad (3)$$

This shows that increasing the individual global objective of each agent can increase the global objective. Therefore,

$$\arg\max_{\theta_i, \forall i} J^G(\theta_1, \theta_2, ...) = \arg\max_{\theta_1, ...} \sum_{i} J_i^G(\theta_i | \theta_1, ...) \quad (4)$$

is hold and maximizing the individual global objective of each agent is equivalent to maximizing the global objective. □

## C  Details and comparison of the driving simulator

In all five environments proposed in this work, we initialize a given number of vehicles in random spawn points, and assign a destination to each of them randomly. Agents are terminated in three situations: reaching the destinations (called *success*), crashing with others or driving out of the road (called *failure*). The new vehicles spawn immediately after existing agents are terminated. Meanwhile, the terminated vehicles remains static for 10 steps (2s) in the original positions, creating impermeable obstacles. Crashing to dead vehicles is also considered as failure. This design enables the total number of vehicles to vary in time and exceed the the initial number of agents, which mimics real-world situation and brings more challenges.

The local observation of each vehicle contains (1) current states such as the steering, heading, velocity and relative distance to boundaries *etc.*, (2) the navigation information that guides the vehicle toward the destination, and (3) the surrounding information encoded by a vector of 72 Lidar-like distance measures of the nearby vehicles. Vehicles are controlled by continuous steering and acceleration signals. The reward function only contains a dense driving reward for longitudinal movement to the destination and a sparse reward that incentives or penalizes the terminations.

MetaDrive [6] is a lightweight and efficient driving simulator implemented based on Panda3D [4] and Bullet Engine. Panda3D is an open-source engine for real-time 3D games, rendering, and simulation. Its well designed rendering capacity enables our simulator to construct realistic monitoring and observational data. Bullet Engine is a physics simulation system that supports advanced collision detection, rigid body motion, and kinematic character controller, which empowers accurate and efficient physics simulation. Empowered by the engine and our optimized implementation, our simulator can achieve high efficiency. In the Roundabout environment, where averagely 40 vehicles runs concurrently, our simulator achieves ∼50 FPS on PC during training. In the less populated environment such as the Parking Lot, where averagely 10 vehicles are running, our simulator can achieves ∼150 FPS. Our simulator will be open-sourced.

Apart from ours, there are lots of existing driving simulators that support RL research. The simulators CARLA [3], GTA V [8], and SUMMIT [1] realistically preserve the appearance of the physical world. For example, CARLA not only provides perception, localization, planning, control modules, and sophisticated kinematics models, but also renders the environment in different lighting conditions, weather and the time of a day shift. Thus the driving agent can be trained and evaluated more thoroughly. Other simulators such as TORCS [16], Duckietown [2] and Highway-env [5] abstract the driving problem to a higher level or provide simplistic scenarios as the environments for the agent to interact with. However, the aforementioned simulators are majorly designed for single-agent scenario, wherein the traffic vehicles are controlled by predefined models or heuristics.

In the MARL context, CityFlow [17] and FLOW [15] are two macroscopic traffic simulators that based on SUMO [7]. However, since these two simulators focus on different aspect of simulating the traffic system, they are not suitable to investigate the detailed behaviors of each learning-based agents.

MACAD [10] provides high-fidelity rendering as the observation based on CARLA. In our preliminary experiments, we have tried to use first-view camera to generate images as the observations, but we find that to be inefficient. So in this paper, we use the scalar states as well as Lidar-like measures as the observation for each agent, which also improves the efficiency of our simulator.

SMARTS [18] is a similar simulator to ours. According to the efficiency test[1], 25 FPS is achieved when 10 agents are running with the scalar state observation as input. However, in our 10-agents environment Parking, our simulator can achieve 150 FPS on single PC even with the Lidar-like observations are feeding to each agent. Besides, our environments cover the major atomic scenes in SMARTS by creating several complex scenes that includes rich driving situations in each scene. For example, in the Tollgate environment, the agents need to learn not only interacting with others, but also interacting with the road infrastructure, the tollgates. They need to learn queueing and patient waiting in the tollgate until being allowed to pass.

The simulator used in [13] is not open-sourced, while our simulator and training code will be available to public. Besides, they only use one scene with relatively fewer vehicles. In [9], the environments are relatively simpler than ours, and the density of traffic are lower. Besides, using the low-level control in our work allowing the RL agents to apply continuous actuation to vehicles directly and learn more diverse behaviors. For example in the Intersection environment, though we do not invite traffic signals as in [9], the negotiation and other social behaviors naturally emerge as the outcome of the proposed CoPO algorithm.

In short, the driving simulator used in this work can run more flexibly and efficiently with complex environments and dense traffic flow.

## D  Implementation details

Table 1 summarizes the hyper parameters used in our experiments.

---

[1]https://github.com/huawei-noah/SMARTS/issues/47

Table 1: Hyper-parameters

| Hyper-parameter | Value |
| --- | --- |
| KL Coefficient | 1.0 |
| $\lambda$ for GAE [12] | 0.95 |
| $\gamma$ for global value estimation | 1.0 |
| $\gamma$ for individual / neighborhood value estimation | 0.99 |
| Environmental steps per training batch | 1024 |
| Number of SGD epochs $K_p$ | 5 |
| SGD mini batch size | 512 |
| Learning Rate | 0.0003 |
| Environmental horizon $T$ | 1000 |
| Neighborhood radius $d_n$ | 10 meters |
| Number of random seeds | 8 |
| Maximal environment steps for each trial | 1M |
| LCF learning rate | 0.0001 |
| LCF number of SGD epochs $K_\Phi$ | 5 |
| LCF distribution initial STD $\phi_\sigma$ | 0.1 |
| LCF distribution initial mean $\phi_\mu$ | 0.0 |

Table 2 summarizes the components used in CoPO. We use three value networks to estimate different cumulative rewards. Note that we do not apply centralized critics since its performance is unstable in SDP-like system due to the variation of local neighborhood. We also leverage a distributional LCF. The reason why we do not use a neural network to predict step-wise $\phi$ is that we consider the local coordination is a episode-level problem. A network predicting step-wise $\phi$ will introduce huge noise in the coordinated objective, which leads to undesired solutions. Finally, since we use parameter sharing [14] among all agents, the extra neural networks introduced by CoPO will not scale up with the increasing number of agents.

| Name | Input Dimension | Output Dimension | Usage |
| --- | --- | --- | --- |
| Policy network | $|\mathcal{O}_i|$ | $|\mathcal{A}_i| \times 2$ | Generate the mean and STD of action distribution to sample actions. |
| Individual value network | $|\mathcal{O}_i|$ | 1 | Predict the value w.r.t. individual reward. |
| Neighborhood value network | $|\mathcal{O}_i|$ | 1 | Predict the value w.r.t. neighborhood reward. |
| Global value network | $|\mathcal{O}_i|$ | 1 | Predict the value w.r.t. global reward. |
| Local coordination factor distribution parameters | 0 | 2 | The trainable $\phi_\mu$ and $\phi_\sigma$ of the social factor. |

Table 2: The neural networks and the model parameters in CoPO.

# E    Learning curves

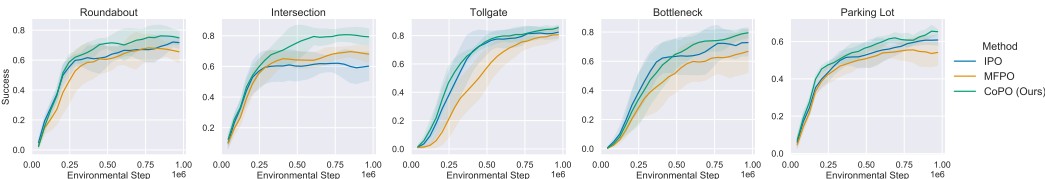

Figure 1: The learning curves of success rate of the populations trained by three approaches.

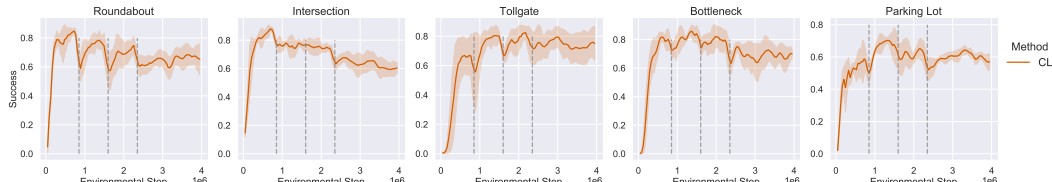

Figure 2: The success rate of populations trained by Curriculum Learning. We split the training into 4 phases with different initial number of agents. The gray lines indicate the time step when we switch the initial number of agents in the environments.

Fig. 1 and Fig. 2 plot the success rate curves of different algorithms. CoPO prevails in all environments. In most of the environment, MFPO shows slower convergence, because learning value function becomes harder as the input dimension is increased.

In curriculum learning baseline, we vary the number of agents to $25\%K$, $50\%K$, $75\%K$ and $100\%K$ at different phases, wherein $K$ is the initial number of agents at each episode. We find that increasing initial agents creates instant drops on the success rate, but the learning algorithm will soon cover. In all the environments except the Tollgate, the performance at the beginning of each phase shows clear downgrade. This makes sense since the tasks in later phase become harder. In Tollgate however, the major challenge is the interaction between agents and the environmental infrastructure the tollgates. This might be the reason why the performance at the final phase is better than the one at the first phase in Tollgate.

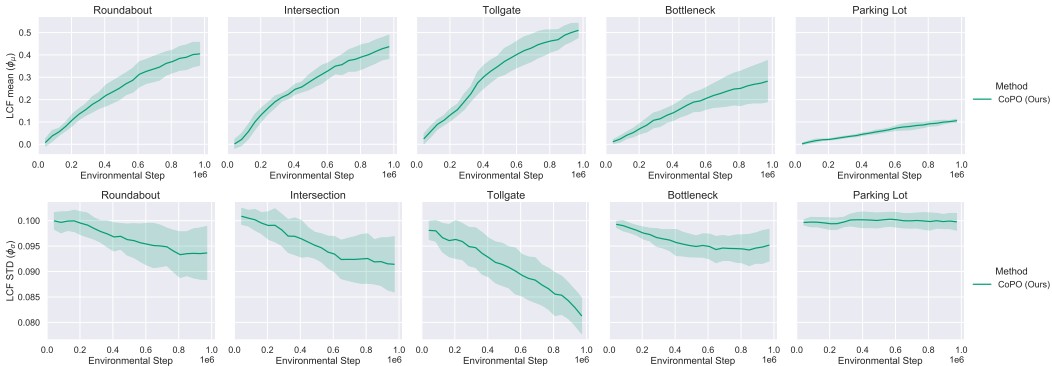

Figure 3: The change of LCF during training.

Fig. 3 demonstrates the varying of $\phi_\mu, \phi_\sigma$ in the course of learning. We find that CoPO tends to increase LCF gradually, and the increasing speed varies across different environments. Meanwhile, the standard deviation decreases, showing that reducing the noise of LCF in CoPO can improve global reward.

## F    Generalization experiments

To justify the generalizability of different training schemes, in test time, we use a simple heuristic to control 25%, 50% and 75% vehicles respectively in the scene. The heuristic we implemented is a rule-based policy which mixes the cruising, lane changing, emergency stopping behaviors with various driving models such as IDM and mobile policy.

In the following table, we can see that introducing heterogeneous policies severely damages the success rate of the IPO population. Noticeably, the CoPO population seems to be mildly affected by IDM policies. This result suggests that the proposed method, due to the distributional LCF in training, can yield more robust policies.

Table 3: generalization-experiments

| Experiment | Roundabout | Intersection | Tollgate | Bottleneck | Parking Lot |
|---|---|---|---|---|---|
| IPO (0% IDM) | 70.81 $\pm$ | 60.47 | 82.90 | 72.43 | 61.05 |
| IPO (25% IDM) | 55.17 | 58.64 | 43.60 | 65.96 | 56.09 |
| IPO (50% IDM) | 52.14 | 57.70 | 42.35 | 62.74 | 54.22 |
| IPO (75% IDM) | 50.49 | 54.35 | 40.19 | 64.86 | 49.69 |
| | | | | | |
| CoPO (0% IDM) | 73.67 | 78.97 | 86.13 | 79.68 | 65.04 |
| CoPO (25% IDM) | 71.55 | 78.40 | 85.72 | 74.99 | 62.60 |
| CoPO (50% IDM) | 63.34 | 77.99 | 85.29 | 69.78 | 58.60 |
| CoPO (75% IDM) | 71.29 | 82.05 | 84.65 | 69.98 | 38.51 |

## G  Social behaviors in the trained vehicle population

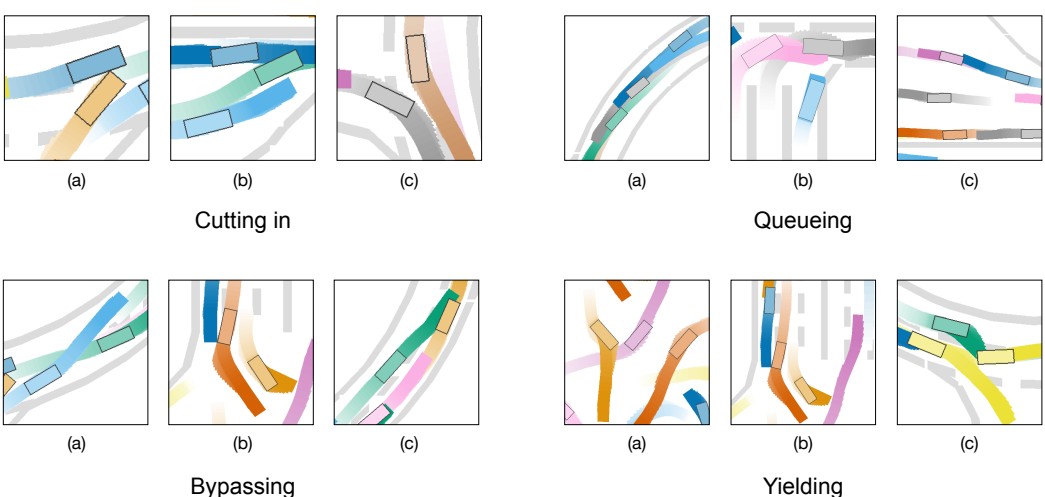

Figure 4: Some social interaction behaviors in the trained population. Following the same visualization method in the main body, we plot the past 25 and future 25 positions of each vehicle with different colors, where the luminosity of a trajectory decreases from light to dark, representing from past positions to future positions.

Fig.4 demonstrates four typical emergent behaviors in CoPO populations: cutting in, queueing, bypassing, and yielding.

Cutting in is a common competitive behavior of human drivers, referring to that one vehicle rapidly rushes into the potential future trajectory of another vehicle in order to take priority. This is a kind of competitive behavior that only benefits the one who conducts it. It can increase the utility of the vehicle that does the cutting in but probably cause unpleasant sudden braking of other vehicles. In (a), the brown vehicle hopes to cut in the path of the blue vehicle, but the blue car sticks to the path. Therefore the crash happens. In (b), the green vehicle cuts in the path of the top blue vehicle, while the top blue vehicle yields to it. In (c), the gray vehicle takes over the original future lane of the brown vehicle because the brown vehicle gives up its path and changes to another lane.

Queueing is another common behavior, where vehicles line up to wait for passing. In (a) and (b), vehicles queue and wait for moving. In (c), which is the upper right corner of the Tollgate environment, vehicles queue for passing, since other vehicles have to stay inside the tollgate for a while.

Bypassing is a behavior that the drivers use another lane different from the current lane and drive fast to bypass some front vehicles. We find CoPO agents learn the bypassing policy to take over some slow front cars. In (b), the middle orange vehicle bypasses the right orange vehicle by slightly

changing its direction. The middle vehicle shifts to its right-hand side a little bit and then rushes rapidly, in order to take over the right vehicle. Both (a) and (c) exhibit excellent lane-changing skills.

Yielding is one important cooperative behavior. The driver stops or even moves toward another direction to spare space for other vehicles to pass. Yielding harms short-term individual interest since the vehicle has to slow down or stop, but it improves long-term reward because crashing to other terminates the episode. The demonstrations in Yielding part of Fig. 4 contain complex interaction behaviors. In (a), three vehicles are negotiating who should go first. Note that we use the term "negotiation", but in fact, no explicit communication is implemented between agents. They have to respond to others only based on their own local observations. In (b), both orange vehicles yield to a pink vehicle. The upper left blue vehicle also yields to the middle orange vehicle when the orange vehicle is bypassing another orange vehicle. In (c), the green vehicle conducts a sudden brake, yielding to a yellow vehicle.

The above demonstrations show the complex social behaviors emerged in the CoPO trained populations. To better illustrate the macroscopic behaviors of each population, we plot the trajectories of each algorithm in each environment as following figures. **Please also refer to the video for dynamic visualization. The video is provided in the supplementary material as well as in the website: `https://decisionforce.github.io/CoPO/`.**

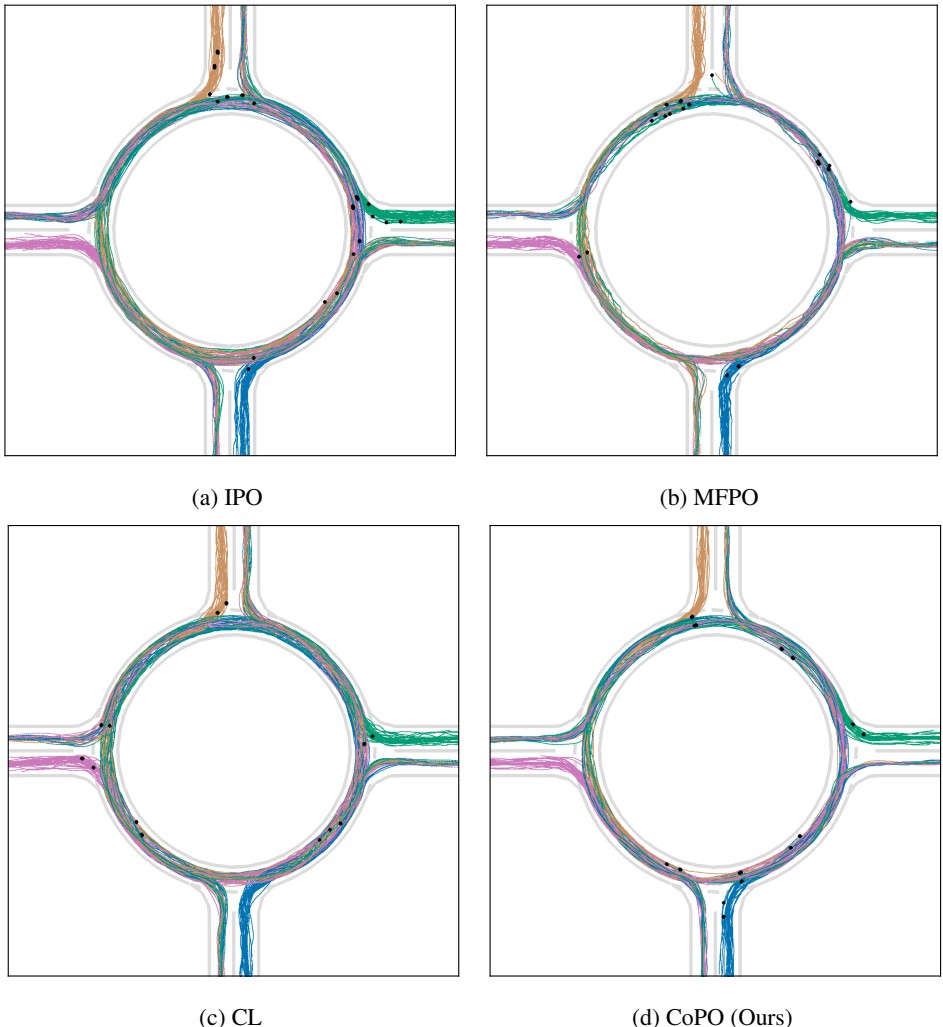

(a) IPO

(b) MFPO

(c) CL

(d) CoPO (Ours)

Figure 5: Plot of trajectories of trained populations in the Roundabout environment.

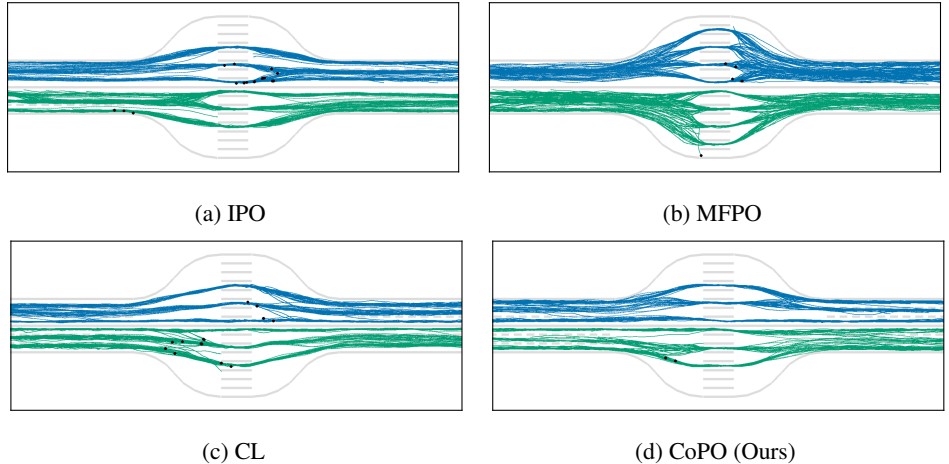

(a) IPO

(b) MFPO

(c) CL

(d) CoPO (Ours)

Figure 6: Plot of trajectories of trained populations in the Tollgate environment.

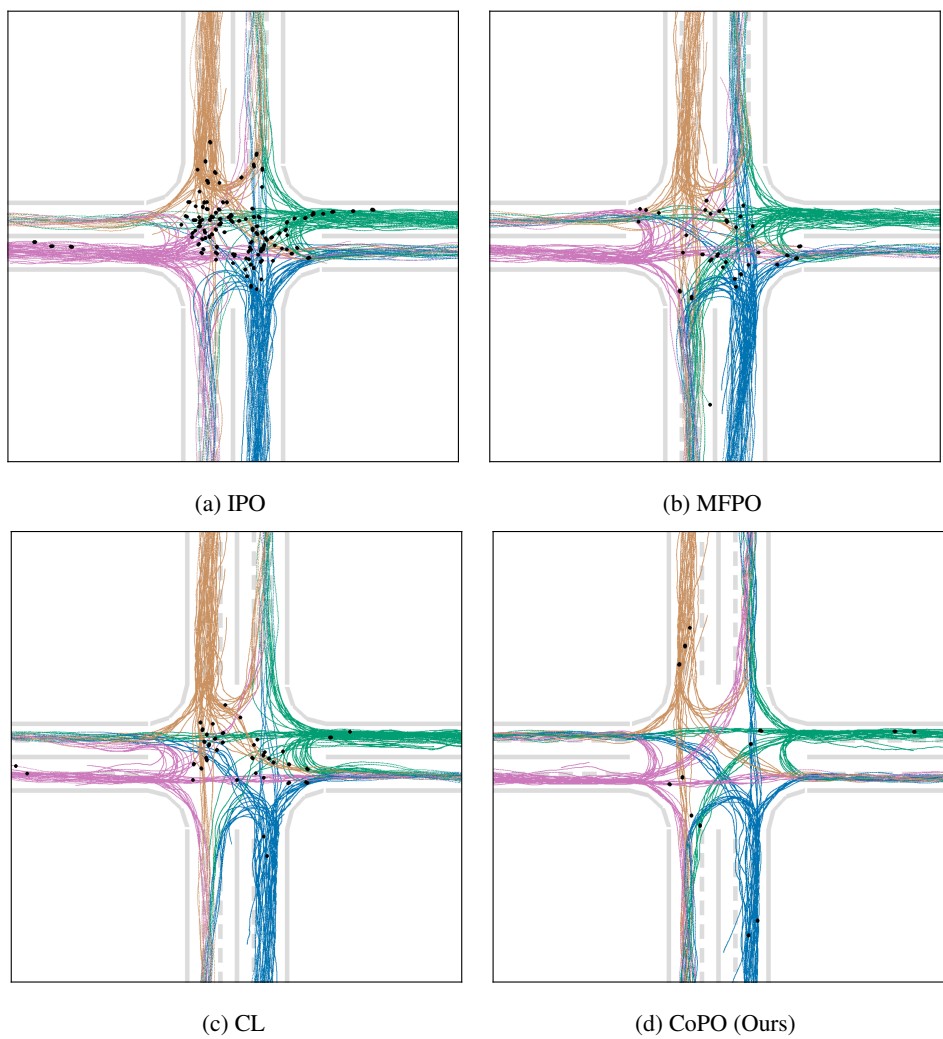

(a) IPO

(b) MFPO

(c) CL

(d) CoPO (Ours)

Figure 7: Plot of trajectories of trained populations in the Intersection environment.

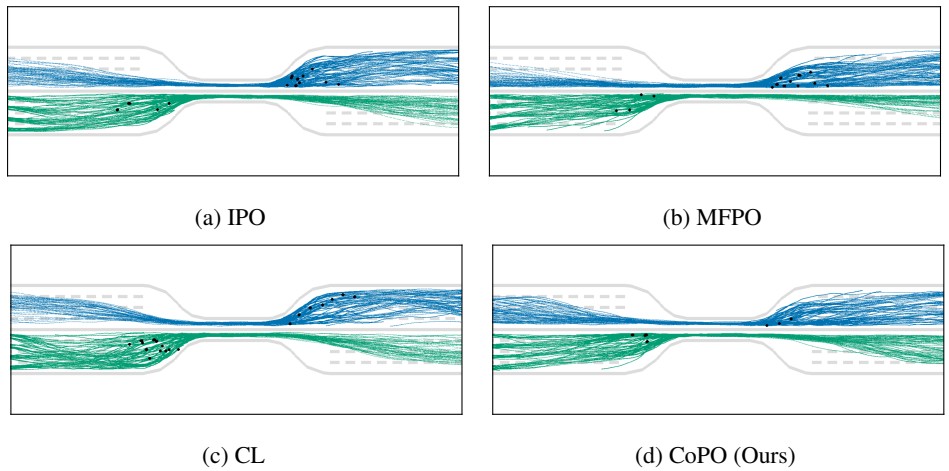

(a) IPO

(b) MFPO

(c) CL

(d) CoPO (Ours)

Figure 8: Plot of trajectories of trained populations in the Bottleneck environment.

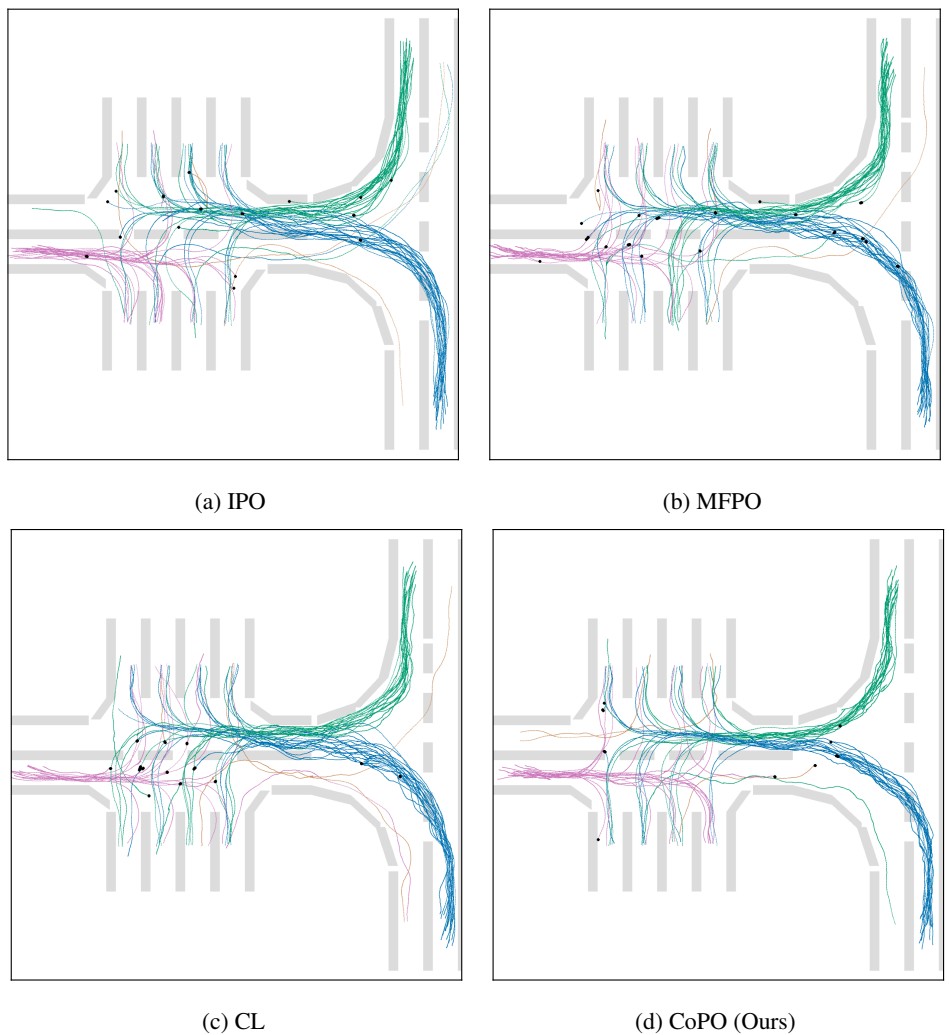

(a) IPO

(b) MFPO

(c) CL

(d) CoPO (Ours)

Figure 9: Plot of trajectories of trained populations in the Parking environment.