# OpenReview forum: "Learning to Simulate Self-driven Particles System with Coordinated Policy Optimization"
_NeurIPS.cc/2021/Conference — NeurIPS 2021 Poster_

### Official Review · Reviewer_xK7J · 2021-07-13

**Rating:** 7
**Confidence:** 4

**Summary:**

In this paper the authors propose a multi-agent reinforcement learning approach to finding controllers in self-driven particles systems. The problem of SDP is formulated as dec-POMDP where each agent has an individual reward function that is wishes to maximize using local observations only.

The algorithm consists of two components: local coordination relying on neighboring rewards and global coordination.

The local coordination part modifies the objective to be a combination of individual rewards and neighbor reward using a Local Coordination Factor. The LCF is used to compute a coordinated objective.

The global coordination component is responsible to find the best LCF, for tractability this objective is factorized into individual global objective.

The algorithm is evaluated on a set of traffic scenarios against multiple baselines: independent policies, mean field methods and curriculum learning.

The reward model using LCF and the split between local and global coordination is sound and interesting. However, the derivation lack of clarity, and it would have been useful to study the algorithm on other domains, especially given that the claim is on general SDP systems.

The authors should revise the claim formulation and provide more justification of the global coordination method.


**Ethical Concerns:**

No strong concern, one could wonder the social implication of the LCF parameters and if this model correctly represent human behavior but this would be future work.

**Limitations And Societal Impact:**

The conclusions of this paper are limited to the traffic domain. It does have societal impact as such solution could help the development of intelligent transportation systems which would have wide impact on our society.

**Main Review:**

The paper is clear and well written and the topic is interesting.

The experiments are focused on traffic scenarios which weakens a bit the claim of the paper. The challenges addressed by the author are valid, it is in practice very difficult to hand-engineer controllers for those systems.  The utility of the solution found by the algorithm needs to be better justified, can we really draw conclusions from the output of this algorithm?

The authors propose a general claim in the introduction, however at the core of the algorithm lies the computation of a neighbor set for each agent. In the traffic domain, neighbors are computed based on distances, how would one compute neighbors in other domain, how is this heuristics affecting the results?

Figure 3: how is success rate defined? It is not clear if the agents are sharing the same LCF, in practice it seems like every individual would have different LCF.

It is understandable that the authors need to factorize the global objective for tractability. However, it is not very clear what the individual global objective represents (the term is contradicting itself), the authors should provide some intuition.

In the proof there seem to be a hidden assumption, a priori, the expectation over trajectories depends on all the agents. Why would the term inside the sum be only depending on \theta_i? This step needs to be clarified.

The experiment show that the proposed algorithm outperforms the baselines on multiple objective on most of the traffic scenarios.

One of the main usage of this training procedure seems to generate a variety of behavior to perform analysis. The interpretation of the generated behavior is very subjective, the author should comment on how to quantitively assess the quality of those behaviors.

In the evaluation procedure, it would be useful to evaluate the generalization of the policy in terms of interaction. What if part of the agent in the environment follows a different policy trained by another algorithm or following a heuristic?

Figure 3 from the appendix should make it in the main paper.


**Time Spent Reviewing:**

1.5

---

> ### Author Response · Authors · 2021-08-10
> **Response to Reviewer xK7J**
>
>
> Thank you for the detailed review! We appreciate your insightful questions which improve this submission. Please see the responses below.
>
> ---
>
> *Q: The utility of the solution found by the algorithm needs to be better justified, can we really draw conclusions from the output of this algorithm?*
>
> A: The key insight of this work is that we can simulate versatile crowd behaviors through our task-agnostic MARL algorithm, which is not limited to traffic simulation.
>
> The only task-specific requirement is to provide a distance metric to measure the neighborhood. The bi-level coordination in CoPO is conducted internally during training and no task-specific component is in there. On the contrary, to achieve coordination, some existing works require access to other agents’ states, actions or dynamics models, or require the information on the road topology to determine the neighborhood. The task-agnostic design makes CoPO applicable to other domains.
>
> ---
>
> *Q: Why do we determine the neighborhood based on distance? How is this heuristics affecting the results?*
>
> A: Using spatial distance to determine neighborhood is a common practice in SDP-like systems such as traffic flow and pedestrian crowd. In other multi-agent domains we can define the features of each agent and measure the neighborhood using the feature difference. We can also use behavior analysis to compute the behavior similarity between agents as the distance. Apart from the specific distance, we can also use K-nearest neighbor (KNN) to define the neighborhood relation. In our preliminary experiment we find that too large neighboring distance yields poor performance (e.g. the distance is so large that all agents are considered neighbors). Smaller distance shows similar results (e.g. $d_n$ = 20m or 10m).
>
> ---
>
> *Q: One of the main usage of this training procedure is to generate a variety of behaviors to perform analysis. The interpretation of the generated behavior is very subjective, the author should comment on how to quantitatively assess the quality of those behaviors.*
>
> A: In the qualitative results we indeed annotate some interesting agent behaviors manually. In the future, we can record the trajectory of each agent in a crowd over time and apply the time-series methods to analyze all the trajectories, such as trajectory clustering and PCA analysis. They will bring more quantitative assessment and comparison of different agent behaviors. We can also compare the trajectories of simulated crowds with real trajectories of traffic scenes collected in the existing trajectory forecasting dataset such as Argoverse.
>
> ---
>
> *Q: In the evaluation procedure, it would be useful to evaluate the generalization of the policy in terms of interaction. What if part of the agent in the environment follows a different policy trained by another algorithm or following a heuristic?*
>
> A: We do this experiment and find interesting results. Please refer to the Q3 in common response for more details.
>
> ---
>
> *Q: The authors should revise the claim formulation and provide more justification of the global coordination method.*
>
> A: In our preliminary experiments, we find that different environments (road topologies) prefer different optimal SVO when we train the population by merely using local coordination with fixed SVO. This introduces extra human supervision and computing requirements. Therefore we introduce global coordination to adjust the reward model through LCF distribution in order to improve the population efficiency. We will improve the writing on global coordination.
>
> ---
>
> *Q: How is success rate defined?*
>
> A: As described in Line 253, the success rate is the ratio of the number of agents that reach their goals over the total occurred agents. Concretely, in one environmental episode, there are many agents that will be spawned. Among the spawned agents, some of them will successfully reach their preset destinations. We compute the success rate of a population by the ratio of the number of successful agents over the number of all spawned agents in one episode. The reported success rate is the average success rate among repeat experiments and those evaluation episodes in each experiment.
>
> ---
>
> *Q: It is not clear if the agents are sharing the same LCF, in practice it seems like every individual would have a different LCF.*
>
> A: Your understanding is correct. Every individual has a different LCF. As described in L5 in the Algorithm 1, each agent will be assigned a LCF sampled from a system-level LCF distribution. We design this mechanism because we wish to introduce more diversity in the population by assigning different LCF to different agents. We will further improve our description in the next version of the paper.
>
> ---
>
> *Q: It is not very clear what the individual global objective represents (the term is contradicting itself), the authors should provide some intuition.*
>
> A: The “individual global objective” is simply the factorization of the “global objective”. We introduce this concept in order to make the optimization of the global objective feasible. We can't optimize the system-level global objective using the data sampled by each individual agent directly. So as a workaround, we prove that by optimizing each individual agent's “individual global objective”, the system-level “global objective” can be optimized. By this way, we can turn the system-level optimization process into an individual agents' optimization process, so that the individual agent's data can be easily used in optimization.
>
> ---
>
> *Q: In the proof there seems to be a hidden assumption, a priori, the expectation over trajectories depends on all the agents. Why would the term inside the sum be only depending on \theta_i? This step needs to be clarified.*
>
> A: Thank you for pointing out this issue in the detailed math proof. Yes, you are right, the corrected equation should be like:
>
> $$\sum_{i} J^{G}_{i}(\theta_1, \theta_2, …,)$$
>
> instead of
>
> $$\sum_{i} J^{G}_{i}(\theta_i)$$
>
> . In the experimental implementation, since we use parameter sharing across all agents, therefore we only have a single policy parameter $\theta = \theta_1 = ... = \theta_i$.

---

> > ### Comment · Reviewer_xK7J · 2021-08-31
> > **Thank you for your reply and for the new experiment**
> >
> > I appreciate that the authors ran a new experiment and carefully answered all of my questions. I invite the authors to add these clarifications and the new experiment at least in the appendix. I recommend this paper for acceptance after reading the rebuttal.

---

> ### Author Response · Authors · 2021-08-31
> **Any post-rebuttal comment?**
>
> Dear Reviewer,
>
> Please kindly let us know if you have any post-rebuttal feedback. We hope our rebuttal could address your concerns and look forwards to hearing back your optimistic acknowledgment of this submission.
>
> Best,
> Authors

---

### Official Review · Reviewer_oVAt · 2021-07-14

**Rating:** 4
**Confidence:** 4

**Summary:**

The authors aim to design controllers for traffic simulation agents in the mixed-motive reinforcement learning setting. They introduce a smooth interpolation between individual and mean global reward (social value orientation) which agents locally optimize while using a meta-optimization process to optimize the degree of interpolation. They show that in a suite of 5 traffic scenarios, their method generally produces agent populations that succeed at higher rates, are more efficient, and produce a smaller amount of critical failures.

**Limitations And Societal Impact:**

There was limited to no discussion of limitations and societal impact.

**Main Review:**

Overall, I enjoyed reading this paper. However, I think it could improve its positioning relative to existing work as well as compare to more related baselines. For these reasons I would suggest rejecting the paper for this conference, but I would look forward to seeing an improved version in a future venue!

In the abstract the authors write "However, previous multi-agent reinforcement learning (MARL) methods define the agents to be teammates or enemies before hand, which fail to capture the essence of SDP where the agents vary to be cooperative or competitive even within one episode." — However, in the introduction and related work the do note some mixed-motive RL works which study exactly this problem. Furthermore, I think the authors missed some key and very related works in their literature review for example "Social Diversity and Social Preferences
in Mixed-Motive Reinforcement Learning" McKee et. al also experiments with social value orientation explicitly. Furthermore, recent works such as Baker et. al 2020, Hughes et. al 2018, Jaques et. al 2019 all work in mixed motive RL in temporally extended games. For these reasons, I would suggest the authors reframe their claims to be more focused on traffic simulations as opposed to general mixed-motive RL. I would also suggest they experiment with some of these prior methods as baselines since they are designed for the same setting.

If I understand correctly, equation 6 is the mean reward of all agents in the episode. If the meta-optimization process is optimizing this value, doesn't that contradict one of the goals of the method which is to not have individuals sacrifice themselves for the good of the whole? I ask because Phi is allowed to go up to 90 (altruistic) so it seems that this would optimize the global reward best. Are there suboptimal local minima the agents fall into when optimizing with phi=90? I think it would be interesting to see Figure 3 but also plotted with equation 6 (global mean reward) as the y-axis in order to understand this better.

Is it possible that there is a different local reward function that would optimize the global efficiency, success rate, and critical failure rate? Is this reward function intractable?

Are there any links to existing game theory literature? It may be a nice addition to this paper to add experiments to smaller more well understood domains to more easily see what types of equilibria it induces (e.g. iterated prisoners dilemma, the coins game, etc).

In the introduction you mention that that your work is unique because it operates in the continuous control space. However, there are many high fidelity robotics works that actually use a discretized action spaces (e.g. see the robotic hand work from OpenAI). Maybe you should consider discretizing the action space and running this baseline if you'd like to make this claim stronger? Or I'd suggest dropping the language about continuous action spaces since I don't think it's a very important part of your work here and maybe distracts from the main point about coordination and social value orientation.

I would be interested to see how results change with varying d_n.

Concatenation is 1 parameterization of a global centralized value function. There are many others (e.g. attention, mean, max, etc). I notice you say that your centralized critic performed worse (L209). Concatenation is not entity invariant, but attention/mean/max are. I would suggest trying these instead.

**Time Spent Reviewing:**

3

---

> ### Author Response · Authors · 2021-08-10
> **Response to Reviewer oVAt**
>
>
> Thank you for the comments and bringing up the papers  on “mixed motive RL” (McKee et. al 2020, Baker et. al 2020, Hughes et. al 2018, Jaques et. al 2019). Indeed those works are relevant and we will add discussions in the related work and tweak the positioning of this work in the intro with respect to those papers. However, we believe our work has three key differences which distinguish our contributions from theirs on mixed motive RL:
>
> **The targeted environments and tasks are different.** We are tackling the continuous control problem in the self-driving/traffic scenario, while the works mentioned mainly focus on grid-world social dilemn environments with discrete action and game-theoretic payoff structure. They are fundamentally different tasks with different complexity and potential output. Thus the developed algorithms in one task cannot be easily adopted to the other task. Besides, our environment usually contains 40 agents in a population setting while the social dilemn environment usually contains only 5 agents. We will open-source our new traffic environments as part of the contributions of this work.
>
> **The focuses are different.** As mentioned by reviewer 5BQc and reviewer xK7J, we aim at introducing the inductive bias to better simulate the SDP used in crowd systems, which is unable to be solved easily by an end-to-end learning method. Therefore our focus is to develop a task-agnostic MARL algorithm to reproduce the emergent collective and socially compliant behaviors of crowds. Instead, their focus is to solve the game-theoretic environments with the insight of social psychology. Both works, which are diagonal to each other, have their own strengths and weaknesses. It will be more promising to see their synergy rather than rivalry.
>
> **The technical details are different.** In this work, we incorporate the social psychological concept SVO into the local coordination, so that we can take neighbors’ utilities into consideration. This is similar to the usage of SVO in McKee et al 2020 (title: Social diversity ...). However, in their work the SVO is imposed as a hyper-parameter and the agents are required to achieve the given SVO. On the contrary, in this work we consider the SVO distribution as an inherent property of the environment, and therefore we propose to use global coordination to adjust SVO automatically. At the agent-level, the agent is still optimizing toward its “coordinated objective”, which is a mixture of native reward and neighborhood reward. Different from McKee et al 2020, Jaques et. al 2019 (title: Social Influence ...) computes an extra social influence reward based on counterfactual reasoning to replace the SVO reward while Hughes et. al 2018 (title: Inequity aversion ...) uses an “inequity aversion reward” that encourages agents to catch up others if other agents have exceeding reward. Our work is different from these two works on two aspects: (1) In reality, it is impractical to access other agents’ actions when computing the action of the target agent, so counterfactual reasoning in Jaques et. al can’t be easily implemented (and we are also using continuous action space which makes this even harder). (2) We propose the global coordination mechanism to dynamically and autonomously adjust the “degree of coordination”. Baker et. al 2020 (Emergent Tool Use ...) tackles a typical mixed cooperative & competitive task where the roles of the agents in their work are fixed within one episode and are not in the varying-role setting we are discussing. Besides, we propose a novel bi-level coordination method to address the specified problem emerging in the SDP-like system, which is not considered in their works.
>
> ---
>
> *Q: Concatenation is one parameterization of a global centralized value function. There are many others (e.g. attention, mean, max, etc). I notice you say that your centralized critic performed worse (L209). Concatenation is not entity invariant, but attention/mean/max are. I would suggest trying these instead.*
>
> A: We already tried this. Please note that the **Mean Field Policy Optimization (MFPO)** method is exactly the "**mean parameterization of global centralized value function**" method you described. We use this as an important baseline in the main body of the paper while showing the result of "concatenation parameterization" in the appendix to justify the usage of MFPO.
>
> ---
>
> *Q: If the meta-optimization process is optimizing Eq. 6, doesn't that contradict one of the goals of the method which is to not have individuals sacrifice themselves for the good of the whole?*
>
> A: No. First of all, global coordination only adjusts the LCF distribution and has implicit impact on policy learning. Second, aligned with Fig. 3, we observed that when the local coordination factor is close to 90 deg, all vehicles become extremely conservative and keep waiting all the time when encountering other vehicles (since they only need to make sure neighbors achieve high reward). Therefore, global coordination will not lead the agent to sacrifice themselves.
>
> ---
>
> *Q: I would be interested to see how results change with varying d_n (the neighborhood range radius).*
>
> A: In our preliminary experiment we find that too large d_n yields poor performance (e.g. d_n is so large that all agents are considered neighbors). But a smaller d_n shows similar results (e.g. d_n = 20m or 10m).
>
> ---
>
> *Q: Why don’t you discretize the continuous action space?*
>
> A: Continuous control environments are fundamentally different to grid world discrete examples. That’s why most of the SOTA RL methods such as SAC and TD3 are evaluated on the continuous control problems in MuJoCo Environments instead of their discretized counterparts. Our goal is to simulate an SDP-like system which has complex continuous actions. More specifically for the driving of a vehicle, steering the wheel, acceleration, brake are all continuous actions in nature, discretizing those actions results in limited behavioral diversity and tuning of the hyperparameters, which lead to suboptimal performance.
>
> ---
>
> *Q: Are there any links to existing game theory literature?*
> A: Yes, we will talk about the relevance to game theory and its potential future work.

---

> > ### Comment · Reviewer_oVAt · 2021-09-01
> > **Score unchanged. Rejection recommended.**
> >
> > Thank you for writing a detailed response and clarifying some details. However, the response with respect to prior mixed-motive RL work makes me feel the authors do not fully understand many of the prior algorithms that have been proposed. I think this paper would be acceptable by either limiting the scope of it's claims to traffic scenarios (and not general mixed-motive MARL), or keep it's claims and run CoPo in prior environments and/or run prior algorithms in their environment. Since the authors have not offered to do either of these but rather argue that they cannot actually be compared to prior mixed-motive work (which is incorrect in my opinion; see below) while still maintaining a claim of generality, I do not change my score and recommend this paper to be rejected.
> >
> > "We are tackling the continuous control problem in the self-driving/traffic scenario, while the works mentioned mainly focus on grid-world social dilemn environments with discrete action and game-theoretic payoff structure. They are fundamentally different tasks with different complexity and potential output. Thus the developed algorithms in one task cannot be easily adopted to the other task." This is incorrect. Policy gradient, the inner loop algorithm you use can be used to optimize both continuous and discrete input and output spaces. In fact, unless I'm grossly misunderstanding your algorithm there is nothing about it that requires a continuous action space or is better suited to one as compared to any other policy gradient method. Your method should be easily applicable to the environments used in those papers, and similarly the methods proposed in those papers could be applied to your environment.
> >
> > In your rebuttal you write "The targeted environments and tasks are different." and also "Therefore our focus is to develop a task-agnostic MARL algorithm". It seems contradictory to claim to have a task agnostic algorithm but also that you focus on different tasks and therefore cannot compare to prior algorithms.

---

> > > ### Author Response · Authors · 2021-09-02
> > > **Clarifying the misunderstanding**
> > >
> > > Thank Reviewer oVAt for the last-min comment on the rebuttal. We want to take this last chance to clarify the misunderstandings of Reviewer oVAt before the discussion phase closes.
> > >
> > > *1. We never claim our method is a general algorithm to solve all the mix-motive RL problems.*
> > >
> > > In the submission, we claim our work at reproducing the collective motions of SDPs. SDPs are population-based systems like traffic and pedestrian crowd.  As summarized more specifically in the rebuttal, we have discussed the differences of our method and the works of mix-motive RL pointed by the reviewer, from the perspectives of different tasks, focuses, and algorithm details. We don't know how to be more clear on this part.
> > >
> > > We are eager to work with the reviewer to update the wordings in the submission that might lead to misunderstanding our claim.
> > >
> > > *2. About "Task-agnostic MARL algorithm...." in the rebuttal.*
> > >
> > > If the reviewer had looked more carefully over the sentence, it wouldn't be contradictory to the claim. The complete sentence is, "Therefore our focus is to develop a task-agnostic MARL algorithm to reproduce the emergent collective and socially compliant behaviors of crowds." Again, here we already narrow the scope of this algorithm to produce the collective crowd movements. The 'task' here means the crowd's specific objective in different scenarios, like going through a roundabout or passing the tollgates. So 'task' here doesn't mean all the tasks in mix-motive RL problems.
> > >
> > > *3. Experiment to evaluate prior works of mix-motive RL on our environment.*
> > >
> > > For this last-min demand, we are working on it. We will keep you posted about the progress and include the result here once we have it.

---

> > > ### Author Response · Authors · 2021-09-04
> > > **New experiment of mixed motive RL method in SDP-like traffic environment**
> > >
> > > We now post our latest experimental result.
> > >
> > > We conduct the experiment using the method proposed in (Jaques et. al 2019) on our SDP environments and find that our proposed method still achieves superior performance. The performance difference, in our humble opinion, is due to the difference in the problem setting between our SDP-like system and the mixed-motive RL.
> > >
> > > ## We run the method (Jaques et. al 2019) in our environment
> > >
> > > We adopted the Modeling Other Agents (MOA) and Social Curiosity Module (SCM) in the open-sourced repo (Jaques et. al 2019) and accommodated the training code to our RL environment. Here is the success rate of different methods on the Intersection environment after 1M steps of training:
> > >
> > > * IPO: Mean 60.47, Std 5.79
> > > * Social Influence (Best variant): Mean 66.14, Std 7.01
> > > * **CoPO (Ours): Mean 78.97, Std 4.23**
> > >
> > > We can see that the proposed CoPO method still achieves better performance with a clear margin compared to the mixed-motive RL baseline requested by the reviewer.
> > >
> > >
> > > ## Why does the Social Influence method achieve not an ideal performance in our environment?
> > >
> > > We hypothesize it is because of the varying neighbors problem as we discussed in the submission and rebuttal.
> > >
> > > The neighborhood of an agent is always changing in different steps. This setting creates two unique features: (1) the total number of agents in the environment is large and varying over time, and (2) the intentions of different agents are different (e.g. some agents go left in the intersection while others might go right).
> > >
> > > Those features bring a challenge for the Model of Other Agents (MOA) proposed in (Jaques et. al 2019). MOA predicts the action distribution of nearby agents by observing the state and other agents’ actions. In the original implementation, the input of MOA is preset to the joint action of all agents. This makes MOA realize the actual agent who provides the actions (though some input dimensions will be masked out if the agent is invisible). However, in our environment, the agents are varying all the time and the MOA can not infer the actual identities as well as the intentions of neighbors. For example, an agent A might get close to the target agent at step T. In later step T+1, agent A goes far from the target agent and another agent B shows up and gets closed. MOA of the target agent will receive the actions in the same input dimensions from two different agents in consecutive steps. Therefore it's hard for MOA to predict the action distributions of neighbors. We believe there exist better ways to encode the information of nearby agents so that the learning of MOA will become easier. But this is out of the scope of current work.
> > >
> > >
> > > Though the relevant method shows sub-optimal performance in our novel SDP-like traffic environment, **the result does not suggest that COPO outperforms in mixed motive RL. These two methods are tackling different problems**. If we can address the varying neighbors problem (e.g. finding a better surrounding representation that can capture neighbors’ identities and intentions) in this SDP-like system, we believe our proposed method COPO can also combine with the social influence method and other important techniques in mixed motive RL. We will include the discussion in the revision.
> > >
> > >
> > > ---
> > >
> > > The following part discusses the technical details in implementation.
> > >
> > > To adopt the previous method to our environment, though it seems that we only need to change the discrete action space into continuous, this actually introduces many new challenges:
> > >
> > > **Why do we run this method?** Among four works (McKee et. al 2020, Baker et. al 2020, Hughes et. al 2018, Jaques et. al 2019) that are pointed out by the Reviewer oVAt, we only find one (Jaques et. al 2019, Social Influence) providing the open-sourced training code at: https://github.com/eugenevinitsky/sequential_social_dilemma_games
> > >
> > > **How do we transfer from discrete action space to continuous action space?**
> > >
> > > Social influence method computes a social influence reward based on the counterfactual action difference. Concretely, the method uses the MOA for each agent to predict other agents’ action distribution on two situations: (1) if the target agent takes the real action as it does in sampling, and (2) if the target agent takes some other actions. The KL divergence between these two predicted distributions is used as the intrinsic reward. However, computing the counterfactual action distribution is hard in continuous action space, since we can’t iterate over all possible target agent’s actions. We change this part by iterating over 9 fixed actions: [steering = -0.5, acceleration = -0.5], …, [steering = +0.5, acceleration = +0.5] (step size of each dim is 0.5).
> > >
> > > When computing the MOA loss, we replace the cross entropy loss between the categorical predicted action distribution and the real action distribution by negative log probability over two Gaussian action distributions.
> > >
> > > The MOA uses the joint action space of all agents as the neural network input. However, in our environment the total number of agents is large and varying. To tackle this problem, we use the nearest 5 agents' actions as input and ask MOA to predict their action distributions.
> > >
> > > **How do we transfer from image-based (grid world) observation to ours?**
> > >
> > > First, we replace all CNN by MLP in feature layers. Besides, we reduce the layers of the network and remove LSTM by MLP. These two choices are based on evaluation results:
> > >
> > > * Social Influence (Large LSTM): Mean 0.1157, Std 0.0053
> > > * Social Influence (Large MLP): Mean 0.2721, Std 0.0032
> > >
> > > We reduce the size of MOA encoder to make sure the policy/value network has same size in our method:
> > >
> > > * Social Influence (Large MLP): Mean 0.2721, Std 0.0032
> > > * Social Influence (Small MLP): Mean 0.6614, Std 0.0700
> > >
> > > We also conduct experiments to verify our transferred code base. We use the same codebase but set the social influence and MOE rewards weights to zero so that the algorithm is supposed to reduce to vanilla PPO. Here is the result:
> > >
> > > * Social Influence (Small MLP):    Mean 0.6614,    Std 0.0700
> > > * [Ablation] Social Influence (Small MLP): Mean 0.6745, Std 0.0303
> > >
> > > * Social Influence (Large MLP):    Mean 0.2721, Std 0.0032
> > > * [Ablation] Social Influence (Large MLP): Mean 0.3230, Std 0.0255
> > >
> > > We hope these newly added empirical results can better reveal the differences between the proposed method designed for SDP and the mixed-motive RL methods proposed for game-theoretic environments.

---

> ### Author Response · Authors · 2021-08-31
> **Any more suggestions or comments?**
>
> Dear Reviewer,
>
> Please kindly let us know if you have any post-rebuttal feedback. It will be great to see our rebuttal can address your concerns to some degree and you would become more optimistic about our submission along with all the other positive reviewers.
>
> Best,
> Authors

---

### Official Review · Reviewer_5BQc · 2021-07-16

**Rating:** 7
**Confidence:** 4

**Summary:**

The authors introduce a novel MARL learning algorithm, CoPO, that is designed to facilitate coordination in SDP systems. In addition to independent value functions, CoPO learns a neighborhood value function according to a return based on the normalized sum of neighboring agent rewards.  Both independent and neighborhood returns are mixed together according to a popular measure of social value orientation. Using a centralized bi-level optimization process, the agent social value distribution is optimised at the same time as the optimal low-level continuous decentralized control policies. The authors evaluate CoPO on a testbed of 5 common road traffic coordination tasks and according to three novel collective metrics. CoPO is shown to outperform other state-of-the-art CTDE MARL algorithms.


**Limitations And Societal Impact:**

It would be great if the authors could connect their findings concerning the suboptimal performance of centralized value functions over decentralized ones in the light of recent developments in the field [Lyu et al 2021].

I would be interested to see what happens if we put a multimodal prior on the Phi-proposal - do we observe optimal multi-modal distributions for agent social value orientations?

Another interesting line of research would be to study mixed autonomous-human traffic. For this purpose, one might want to consider dynamically varying Phis depending on what social value orientation is detected for a given traffic neighborhood.

The authors would do well to check grammar and spelling in isolated instances.

**Main Review:**

The authors' choice of topic fills a gap in MARL literature. While prior work on social value orientation in autonomous driving [e.g. Schwarting 2019] seeks to estimate the participants' social values from observed trajectories, CoPO seems to be the first work that tries to learn optimal social value orientations end-to-end in a CTDE MARL setting. I believe that this represents an important change of perspective: While prior work has focused on social value orientations being inherent properties of human drivers that cannot be influenced by autonomous drivers but have to be anticipated and respected, CoPO is based on the idea that incorporating social value functions in fact yields an effective inductive bias that allows end-to-end MARL methods to effectively solve certain general-sum games (also known as non-zero-sum or mixed cooperative-competitive) for purely autonomous drivers as well. Unlike human drivers, autonomous drivers can tune their "social" value distributions. The fact that this results in such a significant impact on task completion across a range of common traffic coordination problems is, in my view, a surprising insight and validates the original approach of this work.

The submission is written clearly. The bi-level optimization method CoOP is clearly described and well-motivated. Both evaluation environments and metrics used are well described, and experimental results are presented clearly and indicate significant outperformance.

Overall, I recommend this work for acceptance.

**Time Spent Reviewing:**

3

---

> ### Author Response · Authors · 2021-08-10
> **Response to Reviewer 5BQc**
>
> Thank you very much for your praise on this work! As described in the paper, we do believe this work is tackling a new problem in MARL literature. Your understanding that the local coordination imposes effective inductive bias to solve general-sum games is insightful. Please see below the responses to the concerns you raised.
>
> ---
>
> *Q: It would be great if the authors could connect their findings concerning the suboptimal performance of centralized value functions over decentralized ones in the light of recent developments in the field [Lyu et al 2021].*
>
> A: Thank you so much for pointing out this paper Lyu et. al 2021 (title: Contrasting Centralized ...). We believe that work provides the theoretical explanation of the suboptimal performance of the centralized critics methods shown in our experiments. A centralized critic should have a lower bias but higher variance because the policy updates need to be averaged over (potentially many) other agents. This observation is more significant in our evaluation environment where the neighbor agents are constantly changing over time. Therefore a huge variance on the centralized critics eventually damages the training.
>
> ---
>
> *Q: We can use multimodal prior on the SVO (Phi) distribution.*
>
> A: Your suggestion is interesting to explore. Currently we don’t sweep the distribution parameterization of SVO. In our intuition, the optimal SVO distribution is related to the environment structure, thus different map structures have different converged SVO distributions. We find no clues in experiments that there exist multiple optimal SVO distributions for the same map structure (since the results from different repeated experiments are similar).
>
> ---
>
> *Q: Another interesting line of research would be to study mixed autonomous-human traffic. One might want to consider dynamically varying SVO based on neighborhood.*
>
> A: We agree that mixed traffic is an interesting direction. In the Q3 of common responses, we conduct an experiment on testing the trained population in a mixed autonomy scenario. Your idea of introducing human pedestrians is even cooler.  However, we wish to focus this work on generating an SDP-like system (e.g. the socially compliant population of driving agents) in an isolated system with minimal manual design and prior knowledge on the driving models. Varying agent’s Social Value Orientation (SVO) based on the observed SVO of neighbors is an interesting idea. We assume SVO is an inherent property of an agent in this work, but indeed one’s SVO would change in different situations. Inferring others' SVO to adjust one's behavior is an interesting future direction.

---

> > ### Comment · Reviewer_5BQc · 2021-08-29
> > **Reviewer response**
> >
> > Many thanks for addressing my questions. I am happy that my feedback seemed to be useful to the authors.
> > My score remains unchanged.

---

### Official Review · Reviewer_6S1x · 2021-07-16

**Rating:** 8
**Confidence:** 5

**Summary:**

The paper proposes a coordinated policy optimization algorithm to simulate self-driven particles and showcases its power in the autonomous driving application of traffic simulation. The algorithm proposes both local and global coordination mechanisms to tradeoff cooperation and competitiveness at the scene-level. The paper benchmarks the efficiency and safety of their method against a few baselines on a newly proposed simulator of heavily interactive scenarios.

**Limitations And Societal Impact:**

I did not see a discussion about the limitations or societal impact. Please comment on the rebuttal and include this in later revisions.

**Main Review:**

The paper is well motivated from the perspective of the challenges. It is clear that the key challenge in simulating microscopic traffic is the agents' coordination, while things like dynamic feasibility etc. at the agent level are easier. However, I missed a more detailed motivation in terms of potential applications. I think this type of multi-agent simulation (or planning) have multiple applications such as testing/training a separate autonomy stack or the more hypothetical scenario in which all autonomous cars are driving with the same, share autonomy system. While the latter is well covered by the proposed framework, I believe there are no studies or intuition about the suitability of the proposed algorithm on the former. Would be great if the authors can clarify during rebuttal.

In terms of related work, the paper has pretty good coverage but I found 2 rather relevant works missing:
[A] Suo, Simon, et al. "TrafficSim: Learning to Simulate Realistic Multi-Agent Behaviors." Proceedings of the IEEE/CVF Conference on Computer Vision and Pattern Recognition. 2021. --> This is based on Imitation Learning but very focused on the interaction modeling of realistic, scene-level trajectories.
[B] Tang, Yichuan. "Towards learning multi-agent negotiations via self-play." Proceedings of the IEEE/CVF International Conference on Computer Vision Workshops. 2019. --> This one leverages RL and a zoo of models to iteratively achieve more complex and diverse simulated agents.
I would like to see the revision of the paper including a discussion of these, and it would also be great to include them as baselines in the future.

Regarding the method, I believe it is sound and intuitive. While the local coordination is a simple extension of previous approaches, I believe the global coordination mechanism and its learning is original. I like the fact that the proposed method is decentralized at inference despite learning being coordination-aware, and that the local coordination factor is modeled as a continuous degree between cooperation and competitiveness. The write-up is also pretty clear overall. Nonetheless, I have a few questions and comments:
- In the preliminaries, it says that "the policy graddit methods are the appropriate solution", but that seems like a big claim without much backup.
- In Eq. 1, would the intuition that optimizing the individual advantage is hard given that we only observe a single action from the rest of the agents and thus the effect of the agent action cannot be well identified?
- The radial neighborhood is overly simplistic. I believe this is probably the reason why there are still critical failures happening in intersection (as seen in Fig. 6), since the conflicting vehicles come from different branches that are far away.
- I am confused about the statement in L168 about a shared and fixed LCF sinc in Section 4.2 it seems that the LCF is optimized per actor, which makes more sense to me.
- Why is $\phi$ uniformly sampled between [0, 90] (L167) instead of [-90, 90] as stated in L164?
- In Eq. 6, should it be $r_{i, t}^C$ instead of $r_{i, t}$? If not, why?
- In L185, it says that $\gamma$ is removed because the start steps of agents are not aligned. Couldn't the discount factor just go inside the sum over actors? Or do the authors mean it is implemented this way but simplified in the Eq. 6?
- How is the output parameterized, more concretely?

Finally, regarding the experiments, I believe they provide good insights into the difference between the baselines and the proposed method, although more baselines would have made the submission stronger. Aside from the scarcity of baselines, I have the following questions:
- Why are the new traffic environments required? By looking at Figure 1, it looks like a similar environment such as HighwayEnv could be sufficient.
- Since the networks are shared across all agents, I wonder how different goals are encoded for different agents. For instance, in Parking Lot, the agents can navigate toward external roads or enter parking lots. Would be great if the authors could share some details about how this is achieved.
- Why is a basic LiDAR simulated? Is there realistic background? If not, I do not really understand how this contributes to better experiments as opposed to just having access to past tracks of other agents.
- In Fig. 4, why does CoPO achieve less efficiency?
- I missed an experiment where we train a simple planner on the simulated agents for the proposed method and baselines, and evaluate in a common set of scenarios/agents. This would be similar to Table 3 in TrafficSim [A].
- A bigger concern is the claim about greater diversity in the abstract. I believe this is not backed up by any metric or qualitative results. If anything, Fig. 6 suggests that CoPO generates __less__ diverse traffic than the baselines as the lines of traffic flow are "thinner".
- In the video, it seems that most CoPO collisions in roundabout happen before entering it, as opposed to when merging as in most of the cases for the baselines. Any intuitions on why?

Minor comments:
- L101: even if well-known, include the full name of the Dec-POMDP abbreviature.
- In the footnote at the end of page 4, how is the meaningful half range enforced? Clipping?
- L240: sentence is oddly phrased. Perhaps replace "being terminated" for "are terminated".

PS. Figures 6 and 7 are great and the visualizations on the video too! Really enjoyed looking at the results, although would have appreciated narration on the video while highlighting particular cases. Please address as many comments as possible during the rebuttal.

==========

Post rebuttal comments:
- Thanks for the new experiments where IDM is used to control a percentage of the agents. The results are interesting and do show better robustness for CoPO than IPO. However, the results are still concerning in that the effectiveness and applicability of this method seems to be a bit limited to the coordinated agents setting, and the variance of the impact across scenarios seems high. Also, I wanted to make a comment that Tang et. al, does not just use IDM. IDM is used only as an initialization, then a policy is fully learned over multiple training iterations where the other agents policy get picked from an increasing model zoo.
- Thanks for clarifying about the goal encoding. Please include this information in the final revision.
- Overall, the responses and other reviewer comments were aligned with my expectations and thus I maintain my score.


**Time Spent Reviewing:**

6

---

> ### Author Response · Authors · 2021-08-10
> **Response to Reviewer 6S1x**
>
> Thank you very much for providing such a detailed review. We really appreciate the literature you point out and will add them into the next version. Besides, we will rephase some unclear descriptions as you point out. The following are the responses to your questions.
>
> ---
>
> *Q: What would happen if we test/train a separate autonomy stack?*
>
> A: Thank you for the  suggestion. We test the generalization capacity of the different trained populations by controlling part of the vehicles with heuristics similar to the way in “Tang et. al, Towards …”.  Noticeably, we find that the CoPO trained policies are much more robust than baseline. Please refer to the Q3 in common response for more details.
> However, this work focuses on generating a socially compliant population of driving agents in a realistic way with minimal manual designs and prior knowledge on the driving models. Therefore mixed autonomy is not the major focus. That will be part of our future work.
>
> ---
>
> *Q: Why are the new traffic environments required?*
>
> A: Though many prior works propose simulators that support similar road networks and multi-agent traffic, we choose to develop new traffic environments to emphasize the efficiency and extensibility. The new simulator can run at 80 FPS with 30 vehicles in a single process, while other python-based simulators like HighwayEnv or "heavy" simulators like multi-agent CARLA are unable to achieve the same efficiency. It is easy to create new scenarios and road maps in multi-agent settings. Besides, the proposed simulator has a better 3D Bullet kinematics model than the bicycle model used in HighwayEnv. We will open-source the traffic environments for the research community.
>
> ---
>
> *Q: A bigger concern is the claim about greater diversity in the abstract. I believe this is not backed up by any metric or qualitative results.*
>
> A: We use the term diversity since the trained population can produce different human-like behaviors while the baseline populations are full of rushing and clumsy (stochastic) agents.
>
> For example, in the Intersection environment, the IPO population shows messy trajectories, but they are mostly swaying in the straight road which is not realistic. IPO agents rush into the center and crash with each other. On the contrary, CoPO population shows reasonable behaviors: (1) agents are skilled at driving (since they use the optimal tracks) and (2) agents show rich interactions in the road center (as you can see from video and Fig. 7).
>
> In short, the diversity here doesn’t mean the agents in the population should always behave differently, but instead refers to the case that they are capable of demonstrating rich behaviors under different circumstances. Thanks for pointing out this and we will rephrase this part in a later version.
>
> ---
>
> *Q: How different goals are encoded for different agents?*
>
> A: This is an important problem. In each scene, there are many invisible spawn points and destination points. We spawn agents in arbitrary spawn points and assign a random destination point to each of them. After that, we use A* algorithm to find a route from the spawn point to the destination point and split the route into several intermediate points (called checkpoints). The agent receives the relative direction to next checkpoints as part of the observation. We also design an unified reward function: the longitudinal replacement in the route will become the reward. These design choices unify different goals for different agents. For example, in Parking Lot, an agent spawned in an external road will have a destination in parking lots. On the contrary, an agent in a parking lot will have a destination on an external road. The observation and reward however are sharing the same structure in both cases. Thus we have a unified task description that can represent different goals easily.
>
> ---
>
> *Q: Why is a basic LiDAR simulated? Is there a realistic background?*
>
> A: We represent the neighboring information in a relatively low-level manner. Using an ego-centric LiDAR is a common practice in the real world driving systems. To balance the fidelity and the efficiency, we therefore choose to use the one-channel LiDAR as the observation to capture nearby vehicles. Similar to ours, many existing works also choose LiDAR as the observation, such as Hide and Seek (OpenAI), “Emergent Road Rules in Multi-Agent Driving Environments” (ICLR 2021) and so on.
>
> ---
>
> *Q: I missed an experiment where we train a simple planner on the simulated agents for the proposed method and baselines, and evaluate in a common set of scenarios/agents. This would be similar to Table 3 in TrafficSim.*
>
> A: This is an important application if we import real trajectory data from an external dataset. In Table 3 in TrafficSim, the authors do the following steps: (1) they use the real trajectory data to train the trajectory generator TrafficSim, (2) they use TrafficSim to augment original dataset, (3) they use the augmented dataset to train the planner, (4) they compare the planner produced trajectories with original dataset and report the results. As you can see from this procedure, we can’t reproduce the similar evaluation in this work, since currently our simulator doesn’t take a real-world trajectory dataset. But importing real data is a promising direction we would like to pursue in the future.
>
> ---
>
> *Q: I am confused about the statement in L168 about a shared and fixed LCF since in Section 4.2 it seems that the LCF is optimized per actor, which makes more sense to me.*
>
> A: Please note that the L168 statement on LCF is only applicable to the toy experiment in Fig. 3. In this experiment, during the training, we assign each spawn agent with an uniformly sampled LCF in [0 deg, 90 deg]. During the test time, we assign a given LCF, e.g. 0 deg, 10 deg, …, to all agents. We repeat this kind of evaluation multiple times, each time using unique LCFs and collecting the test success rate of the population. By doing so, we can plot a curve as in Fig. 3 showing the relationship between test success rate and test-time LCF.
>
> In the global coordination (Sec 4.2), there is only a singular LCF distribution in the whole system. The Eq. 11 only says that we use the data from different agents to train the same LCF distribution (that's why LCF in Eq.11 does not have any subscription). In short, the LCF distribution is not optimized per agent. The major reason is that the number of agents are always varying in different episodes and maintaining a system-level LCF distribution and instantiating a LCF to each spawn agent would be the most practical implementation.
>
> ---
>
> *Q: In Eq. 6, should it be r_{i,t}^{C} instead of r_{i,t}? If not, why?*
>
> A: We wish to adjust LCF to make the population more efficient, so we choose to use the sum of all native rewards produced by the environment from all agents as the objective of global coordination. If we use the coordinated reward r^{C}, which is the combination of the native reward and the neighborhood reward, then the native reward r_{i, t} will be accumulated multiple times, which deviates from our intuition on the global coordination objective.
>
> ---
>
> *Q: In L185, it says that γ is removed because the start steps of agents are not aligned. Couldn't the discount factor just go inside the sum over actors?*
>
> A: First of all, the global coordination objective Eq. 6 considers the sum over all emerging agents in one episode. So it is meaningless to set a time step as the “anchor” and accumulate all agents’ rewards to that “anchor”. Because doing so will discriminate against the agents that spawn in later time far from the “anchor” in the global objective (their rewards are more discounted).
>
> ---
>
> *Q: How is the output parameterized, more concretely?*
>
> A: We don't exactly grasp what "output" means here. For the Gaussian policy we used, the output vector can be decomposed into two parts: the mean and the log STD of the action distribution at the current state. For the value networks, the output is a scalar that stands for the expected return. For the LCF, we use two learnable parameters to serve the mean and log STD of the LCF distribution (so LCF distribution is invariant to the current time step and current state, it is in system-level, instead of in step-level).
>
> ---
>
> *Q: In Fig. 4, why does CoPO achieve less efficiency?*
>
> A: As shown in Fig. 4, in the Tollgate environment, the efficiency of CoPO is inferior compared to other baselines. Tollgate environment has an unique feature: passing the tollgate will yield “passing reward” to the agent. We hypothesize this feature leads to interesting exploitation of the CoPO algorithm: the agent learns to drive slowly after it passes the tollgate, so that it can receive the “passing reward” of later agents in the “neighborhood reward”. This makes the velocity of the CoPO population lower than others, leading to less efficiency.
>
> ---
>
> *Q: In the footnote at the end of page 4, how is the meaningful half range enforced? Clipping?*
>
> A: We use the LCF as an additional observation dimension. Therefore we normalize it with "clip(LCF, 0, 90) / 90". When sampling a LCF from the LCF distribution, we clip the sampled value by "clip(LCF, 0, 90)”, wherein LCF is sampled from LCF distribution.
>
> ---
>
> *Q: In Eq. 1, would the intuition that optimizing the individual advantage is hard (given that we only observe a single action from the rest of the agents) and thus the effect of the agent action cannot be well identified?*
>
> A: Exactly. Optimizing one's own self-interested reward in the MARL environment is extremely non-stationary, especially in the context of partially observability. We believe that explains a bit on why our bi-level coordination can solve this problem better.

---

### Author Response · Authors · 2021-08-10
**Common Response to Reviewers**

We thank all reviewers for the insightful reviews! Apart from the specific responses to each reviewer, here we address the common issues raised in the reviews on social impact and limitations of this work as well as a new experimental result on mixed policies.

---

*Q1: What is the social impact of the proposed method?*

A1: The proposed method can simulate diverse and complex behaviors of traffic. As pointed out by the reviewer xK7J, this work could help the development of intelligent transportation systems which would have a wide impact on society. We can analyze the emergent behaviors of the traffic under different scene structures and optimize the road structure or traffic light control signals to improve the efficiency of the traffic flow based on those analyses. Moreover, this work can generate rich interactive environments, such as generating diverse traffic scenarios in a driving simulator, which can be used to benchmark the generalizability of the autonomous driving systems. Besides simulating traffic flow, this proposed method is applicable for pedestrian crowd simulation to study social crowd behaviors as well as potential human stampedes and crushes. We will add the discussion of the social impact in the next version.

---

*Q2:  What are the limitations of this work?*

A2: This work focuses on the multi-agent decision-making problem, so we simplify the perception of the vehicles as  one-channel LiDAR and assume the acquisition of accurate sensory input data. In reality, the accurate perception of the surroundings in self-driving remains very challenging. Apart from the simplified perception, this work adopts 5 typical traffic scenarios in a simulator as the testbed. They are still far from emulating the complexity of real-world traffic scenes. In the future we will import real-world HD maps in our simulator to create more realistic and diverse scenarios.

---

*Q3: What will happen if we control part of the agents in the population by other policies?*

A3: We thank reviewer 6S1x and reviewer xK7J for this idea. To justify the generalizability of different training schemes, in test time, we use a simple heuristic (developed in HighwayEnv simulator) to control 25%, 50% and 75% vehicles respectively in the scene. In the following table, we can see that introducing heterogeneous policies severely damages the success rate of the IPO population. Noticeably, the CoPO population seems to be mildly affected by IDM policies. This result suggests that the proposed method, due to the distributional SVO in training, can yield more robust policies.



**Table: The success rate of trained population when partial traffic is controlled by heuristic.**

|                    | **Roundabout** | **Intersection** | **Tollgate** | **Bottleneck** | **Parking Lot** |
| ----------------------- | -------------- | ---------------- | ------------ | -------------- | --------------- |
| **IPO (0% IDM)** | 70.81 | 60.47 | 82.90 | 72.43 | 61.05 |
| **IPO (25% IDM)**       | 55.17     | 58.64       | 43.60   | 65.96     | 56.09      |
| **IPO (50% IDM)**       | 52.14     | 57.70       | 42.35   | 62.74     | 54.22     |
| **IPO (75% IDM)**       | 50.49     | 54.35       | 40.19   | 64.86    | 49.69    |
|                    |                |                  |              |                |                 |
| **CoPO (0% IDM)** | 73.67 | 78.97 | 86.13 | 79.68 | 65.04 |
| **CoPO (25% IDM)**      | 71.55     | 78.40       | 85.72   | 74.99     | 62.60      |
| **CoPO (50% IDM)**      | 63.34     | 77.99       | 85.29   | 69.78     | 58.60     |
| **CoPO (75% IDM)**      | 71.29     | 82.05       | 84.65   | 69.98  | 38.51     |

---

### Decision · Program_Chairs · 2021-09-27

**Decision:**

Accept (Poster)

**Comment:**

The authors propose a method that uses bi-level optimization to overcome social dilemmas amongst large groups of self-interested agents. Specifically, each agent considers a weighted sum of its own reward and the reward of its neighbours. The weighting term, phi, trading off between these two is optimized using meta-gradients of the global objective.

Importantly, by assuming that the method can undertake reward-shaping for the individual agents, the actual social dilemma is no longer relevant and the problem can be considered fully cooperative.


There is a clear divide between the reviewers: While 3 of the reviewers provided very positive reviews, one reviewer remained critical. While the authors addressed the concerns and provided additional (favorable) experimental comparison at short notice, this did not change the score.


Given the positive reviews, the discussion and the overall contributions of the paper, I recommend this paper to be accepted.

However, here are two requests: The problem is currently introduced as a Dec-POMDP. This is inaccurate since different agents have different reward functions.

Furthermore, two necessary baselines need to be added: 1)  Comparison to a fully cooperative setting where the global reward (sum of all agents’ reward) is maximized. 2) Each agent maximizes the total local reward (i.e. neighbors’ and own reward). This is very close to phi=90 degrees but includes the agent’s own reward.